# Broad Spectrum Functional Activity of Structurally Related Monoanionic Au(III) Bis(Dithiolene) Complexes

**DOI:** 10.3390/ijms23137146

**Published:** 2022-06-27

**Authors:** Yann Le Gal, Agathe Filatre-Furcate, Dominique Lorcy, Olivier Jeannin, Thierry Roisnel, Vincent Dorcet, Diana Fontinha, Denise Francisco, Miguel Prudêncio, Marta Martins, Catarina Soeiro, Sílvia A. Sousa, Jorge H. Leitão, Tânia S. Morais, Inês Bártolo, Nuno Taveira, Joana F. Guerreiro, Fernanda Marques

**Affiliations:** 1CNRS, ISCR (Institut des Sciences Chimiques de Rennes)—UMR 6226, Université Rennes, F-35000 Rennes, France; yann.le-gal@univ-rennes1.fr (Y.L.G.); agathe.filatre@gmail.com (A.F.-F.); olivier.jeannin@univ-rennes1.fr (O.J.); thierry.roisnel@univ-rennes1.fr (T.R.); vincent.dorcet@univ-rennes1.fr (V.D.); 2Instituto de Medicina Molecular João Lobo Antunes, Faculdade de Medicina, Universidade de Lisboa, Avenida Professor Egas Moniz, 1649-028 Lisboa, Portugal; dfontinha@medicina.ulisboa.pt (D.F.); denisefrancisco@medicina.ulisboa.pt (D.F.); mprudencio@medicina.ulisboa.pt (M.P.); marta.martins@medicina.ulisboa.pt (M.M.); 3iBB-Institute for Bioengineering and Biosciences, Departmento de Bioengenharia, Instituto Superior Técnico, Universidade de Lisboa, Av. Rovisco Pais, 1049-001 Lisboa, Portugal; catarina.soeiro25@gmail.com (C.S.); sousasilvia@tecnico.ulisboa.pt (S.A.S.); jorgeleitao@tecnico.ulisboa.pt (J.H.L.); 4Associate Laboratory, i4HB—Institute for Health and Bioeconomy at Instituto Superior Técnico, Universidade de Lisboa, Av. Rovisco Pais, 1049-001 Lisboa, Portugal; 5Centro de Química Estrutural and Departamento de Química e Bioquímica, Faculdade de Ciências, Universidade de Lisboa, Campo Grande, 1749-016 Lisboa, Portugal; tsmorais@fc.ul.pt; 6iMed.ULisboa, Faculdade de Farmácia da Universidade de Lisboa, Avenida das Forças Armadas, 1600-083 Lisboa, Portugal; ibartolo@ff.ulisboa.pt (I.B.); ntaveira@ff.ulisboa.pt (N.T.); 7Centro de Ciências e Tecnologias Nucleares and Departamento de Ciências e Tecnologias Nucleares, Instituto Superior Técnico, Universidade de Lisboa, Estrada Nacional 10 (km 139.7), 2695-066 Bobadela, Portugal; joanaguerreiro@ctn.tecnico.ulisboa.pt

**Keywords:** gold bis(dithiolene) complexes, diselenolene, structural modification, anticancer activity, antimicrobial activity, HSA interaction

## Abstract

The biological properties of sixteen structurally related monoanionic gold (III) bis(dithiolene/diselenolene) complexes were evaluated. The complexes differ in the nature of the heteroatom connected to the gold atom (AuS for dithiolene, AuSe for diselenolene), the substituent on the nitrogen atom of the thiazoline ring (Me, Et, Pr, iPr and Bu), the nature of the exocyclic atom or group of atoms (O, S, Se, C(CN)_2_) and the counter-ion (Ph_4_P^+^ or Et_4_N^+^). The anticancer and antimicrobial activities of all the complexes were investigated, while the anti-HIV activity was evaluated only for selected complexes. Most complexes showed relevant anticancer activities against Cisplatin-sensitive and Cisplatin-resistant ovarian cancer cells A2780 and OVCAR8, respectively. After 48 h of incubation, the IC_50_ values ranged from 0.1–8 µM (A2780) and 0.8–29 µM (OVCAR8). The complexes with the Ph_4_P^+^ ([**P**]) counter-ion are in general more active than their Et_4_N^+^ ([**N**]) analogues, presenting IC_50_ values in the same order of magnitude or even lower than Auranofin. Studies in the zebrafish embryo model further showed that, despite their marked anticancer effect, the complexes with [**P**] counter-ion exhibited low in vivo toxicity. In general, the exocyclic exchange of sulfur by oxygen or ylidenemalononitrile (C(CN)_2_) enhanced the compounds toxicity. Most complexes containing the [**P**] counter ion exhibited exceptional antiplasmodial activity against the *Plasmodium berghei* parasite liver stages, with submicromolar IC_50_ values ranging from 400–700 nM. In contrast, antibacterial/fungi activities were highest for most complexes with the [**N**] counter-ion. Auranofin and two selected complexes [**P**][AuSBu(=S)] and [**P**][AuSEt(=S)] did not present anti-HIV activity in TZM-bl cells. Mechanistic studies for selected complexes support the idea that thioredoxin reductase, but not DNA, is a possible target for some of these complexes. The complexes [**P**] [AuSBu(=S)], [**P**] [AuSEt(=S)], [**P**] [AuSEt(=Se)] and [**P**] [AuSeiPr(=S)] displayed a strong quenching of the fluorescence intensity of human serum albumin (HSA), which indicates a strong interaction with this protein. Overall, the results highlight the promising biological activities of these complexes, warranting their further evaluation as future drug candidates with clinical applicability.

## 1. Introduction

Most of the drugs currently in clinical use to treat cancer, microbial or viral infections present several disadvantages, such as their low selectivity, acquired drug resistance, toxicity and severe side-effects, evidencing the need for the development of novel and more efficient strategies. Metal-based complexes can mitigate most of these disadvantages. In fact, metal complexes have attracted considerable interest mainly due to their versatile electronic and structural features that can be explored for rational drug design. The metal ion, its oxidation state, coordination numbers and geometries and type and number of ligands are some of the key features that must be considered in the design and synthesis of prospective drugs for medical use. In line with this, numerous compounds of different transition metal ions are currently under development or reaching the clinical settings for medical applications [1,2,3,4]. Their unique properties offer novel chemistry and provide great opportunities for the discovery of drugs with new mechanisms of action [1,5,6]. In particular, Cisplatin, the well-known platinum drug with a long history in cancer treatment and the knowledge of its side effects, systemic toxicity, mechanism of action, and tumor resistance, has prompted several efforts to explore novel strategies and formulae to develop new metal complexes that exhibit non-classical modes of action with improved balance of risk–benefits [7,8,9,10]. In this context, gold complexes have gained much attention and endured extensive research in the medicinal chemistry domain for the treatment of cancer, inflammatory disorders and infectious diseases [11].

The investigation on the anticancer properties of gold complexes was primarily focused on Auranofin, a gold (I) phosphine derivative that has been investigated as a potential treatment for cancer (e.g., leukemia, ovarian), neurodegenerative disorders, HIV/AIDS, as well as parasitic and bacterial infections [10]. Motivated by the broad spectrum of activity of Auranofin, a diversity of gold(I) complexes bearing phosphine ligands such as triphenylphosphine, diphenylphosphino-alkanes and dithiocarbamates have been synthetized and showed potent antiproliferative activity on a variety of cancer cells [12,13]. On the other hand, gold (III) complexes have been studied as potential anticancer agents due to their square-planar geometry, similar to that of Cisplatin [14,15]. In general, gold (I/III) complexes constitute a diverse family of very promising agents for cancer therapy. Indeed, several gold (I/III) complexes have shown outstanding in vitro antiproliferative activities against several types of cancer cells, in some cases opening new prospects for further advanced preclinical studies. Relevant examples are gold(I/III) complexes with polydentate nitrogen donor ligands, (C^N)-cyclometalated ligands and N-heterocyclic carbenes (NHCs) [16,17,18,19]. Although some gold-NHC derivatives undergo ligand rearrangement reactions in the presence of water, some other complexes are stable in aqueous environment and showed anticancer activity superior to Cisplatin, which makes these complexes excellent potential chemotherapeutics [20,21,22,23,24,25].

The mechanisms underlying the antiproliferative activities of gold(I/III) complexes remain unclear [26,27]. In vitro studies showed that Auranofin can overcome Cisplatin resistance in human ovarian cancer cells, which indicates a mode of action different from the one described for Cisplatin. In fact, Auranofin has been identified as a potent inhibitor of thioredoxin reductase (TrxR), due to the high affinity of gold(I) for the selenocysteine residue of TrxR, thus causing an alteration in the cellular redox balance and a cascade of severe metabolic impairments that result in apoptotic cell death. In some cases, the apoptotic response is mediated by mitochondrial membrane potential depolarization and enhanced caspase activity [19,26,27,28,29,30]. In several cancer cells, high expression levels of TrxR are related to cell growth and proliferation and poor prognosis in a variety of cancers [31].

Interactions with cysteine- or selenocysteine-containing enzymes other than thioredoxin reductase was also reported, such as phosphatases, cathepsin and topoisomerase I [32,33].

Literature examples evidence a DNA-independent mechanism of action for some gold(I/III) complexes, although others report DNA binding, and more recently the binding to DNA secondary structures such as G-quadruplex (G4) nucleic acids. Metal complexes as small G4-ligands are expected to interact selectively with G4 nucleic acids, acting as a promising strategy for the development of anticancer drugs with selective toxicity towards cancer cells over normal ones [34].

The antimicrobial activity of gold(I/III) complexes has not been investigated as extensively as their anticancer activity. Only recently, due to the threat of multidrug-resistant (MDR) pathogens, the application of these compounds as antimicrobials has gained more attention [35]. Nevertheless, a considerable number of gold(I) and gold(III) complexes, mostly complexes with a + 1 oxidation state, have been tested against a broad spectrum of bacteria, fungi and parasites. In general, gold(I) complexes with phosphine showed relevant antibacterial and antifungal activities. Auranofin and gold(I/III) complexes with NHC ligands display antimicrobial potential against many Gram-positive bacteria, including multidrug resistant strains, but poor activity towards Gram-negative species. Gold(I/III) complexes with NHCs ligands were found as efficient inhibitors of bacterial TrxR, indicating that inhibition of this enzyme might be involved in their mechanism of action [36].

The activity of Auranofin against different pathogenic fungi, including multidrug-resistant *Candida albicans*, was also demonstrated [37,38,39,40].

So far, only a few gold(I/III) complexes have been evaluated as antimalarial compounds. Auranofin was identified in a high-throughput screening as having activity against the hepatic stage of the rodent *Plasmodium berghei* parasite in vitro. Furthermore, it displayed in vitro activity against the blood stage of the 3D7 and Dd2 strains of *P. falciparum* [41].

Promising candidates for future development have been the gold(I/III) complexes containing thiosemicarbazonato, phosphines and NHCs ligands [22,42,43,44,45]. The presence of N-containing heterocycles in the series of mononuclear gold(I) complexes is of great importance to the antiplasmodial activity, which is probably related to lipophilicity, basicity, and structural features of these complexes. However, further studies are needed to extensively explore these complexes as antimalarial drugs. The mechanism of action seemed to involve the inhibition of antioxidant systems that are critical for the parasite’s survival. In vitro, these complexes have been shown to directly inhibit the parasite’s TrxR [21].

Whereas extensive work has been conducted with gold(I) complexes, the high oxidation state counterparts gold(III) have not been explored as much. In view of bridging the gap and making a contribution to explore other relevant gold(III) complexes, our group recently demonstrate the potential activity of monoanionic gold(III) bis(dithiolene) complexes against ovarian cancer cells, Gram-positive bacteria, *Candida* strains and *P. berghei* [46,47]. The interactions of these gold(III) bis(dithiolene) complexes with DNA, the classical cellular target of Cisplatin, are negligible, as was observed for Auranofin, suggesting a different mechanism of action for the observed biological effects [46]. In addition, the results obtained indicate that TrxR could constitute a potential target of gold(III) bis(dithiolene) complexes [47]. These promising results prompted us to herein explore the multifunctional therapeutic potential of a novel series of structurally related mononuclear gold(III) bis(dithiolene) complexes as prospective anticancer, antibacterial, antiplasmodial and anti-HIV drugs. Complexes differing from the ones previously described in [47] by the substituent on the nitrogen atom of the thiazoline ring, the nature of the chalcogen atoms within the metallacycles, sulfur vs. selenium, together with the exocyclic sulfur replacement and the counter-ion, Ph_4_P^+^ or Et_4_N^+^ ([Fig ijms-23-07146-ch001]). The effect of modifications of the ligand skeleton and counter-ion on the biological properties of theses complexes will be discussed.

One of the crucial steps in the development of novel drugs is studying the interaction to albumin, the main blood protein. Surprisingly, studies on the interaction of gold(III) complexes with serum proteins have been poorly explored, despite of their extreme relevance for the mechanism of action, toxic effects, as well as their transport and distribution in vivo [15,48,49]. In the present study, we investigated the interaction of selected complexes with human serum albumin (HSA) by fluorescence spectroscopy, taking advantage of the intrinsic fluorescence of this protein.

## 2. Results and Discussion

### 2.1. Synthesis and Redox Properties of the Gold Complexes

The general synthetic strategy used to prepare the different gold monoanionic complexes starting from the N-alkyl-1,3-thiazoline-2-chalcogenone **1-R(=Y)** is outlined in Figure 1. The first step corresponds to the metalation of the heterocyclic ring with lithium diisopropylamide (LDA) followed by the addition of sulfur or selenium and bromopropionitrile, in order to form the protected dithiolene or diselenolene ligand, **2-XR(=Y)** [50]. Then, the deprotection of the dithiolene (X = S) or diselenolene (X = Se) ligand was performed in the presence of sodium methoxide, NaOMe, followed by the subsequent addition of KAuCl_4_.H_2_O and Ph_4_PCl or Et_4_NBr to afford the monoanionic gold complexes, [C][AuXR(=Y)]. After recrystallization in acetonitrile, the gold complexes were isolated as dark greenish crystalline solids. The newly synthesized complexes were all characterized using ^1^H NMR, HRMS, elemental analysis and cyclic voltammetry. The sixteen gold complexes prepared and investigated in this study are collected in [Fig ijms-23-07146-ch001].

The redox properties of the various gold(III) bis(dithiolene) and bis(diselenolene) complexes were investigated by cyclic voltammetry using the same conditions, dichloromethane solutions and NBu_4_PF_6_ as supporting electrolyte. The oxidation potentials are shown in Table 1. Under these conditions, the nature of the counter-ion, Ph_4_P^+^ or Et_4_N^+^, had no influence on the redox properties observed, due to ion exchange with the Bu_4_N^+^ of the supporting electrolyte. Thus, only the result of one monoanion is included in Table 1, even if the two salts were analyzed. For all the investigated monoanionic bis(dithiolene) complexes, three redox processes are observed on the cyclic voltammograms. On the anodic scan, the two oxidation waves correspond to the oxidation of the monoanionic species into the neutral and to the monocationic complex. Most of these complexes had the tendency to deposit on the electrode once oxidized and frequently a strong desorption peak was observed on the reversible scan. This phenomenon occurs essentially for the complex with an exocyclic sulfur atom, [AuSR(=S)]^−1^. On the cathodic scan, an irreversible process associated with the reduction in the monoanion into the dianion occurs (Figure 2). Comparison of the redox potentials shows that when the nature of the side chain was modified from Me, Et, Pr to Bu, no significant differences were observed on the redox potentials. However, the nature of the exocyclic substituent = O, = S, = Se to = C(CN)_2_ significantly affected the redox potentials. The presence of oxygen instead of sulfur makes the oxidation of the monoanionic into the neutral species easier. On the other hand, the presence of electron-withdrawing substituents such as =C(CN)_2_ induces an anodic shift of the oxidation potentials by 300 mV, compared with the =O derivative, leading to a monoanion which is more difficult to oxidize than the other complexes of the series. Both derivatives, [AuSEt(=C(CN)_2_)]^−1^ and [AuSEt(=O)]^−1^ exhibited a wider potential stability window of the neutral species than the sulfur analogues [AuSR(=S)]^−1^ [51]. Indeed, the potential difference between the second and the first oxidation potentials amounts to 470 mV and 410 mV for the =O and the =C(CN)_2_ derivatives, respectively, while for the =S analogues this value is smaller (170 mV). The gold complexes with selenium atoms, either the bis(dithiolene) with exocyclic selenium atoms or the bis(diselenolene) complexes, exhibited a different behavior, since on the first anodic scan an irreversible oxidation process is observed on the cyclic voltammograms presumably due to strong adsorption phenomenon. For comparison, we also investigated the redox behavior of Auranofin using the same experimental conditions in CH_2_Cl_2_ in the presence of NBu_4_PF_6_ as a supporting electrolyte. Using these conditions, upon anodic scan only one irreversible oxidation process is observed at 1.42 V vs. SCE. Therefore, all these monoanionic gold bis(dithiolene) and bis(diselenolene) complexes are easier to oxidize than Auranofin.

### 2.2. Molecular Structure and Intermolecular Interactions in the Solid State

X-ray analyses of novel monoanionic dithiolene complexes [AuSBu(C=S)]^−^^1^, [AuSEt(=C(CN)_2_)]^−1^, [AuSEt(=Se)]^−1^ and [AuSPr(=Se)]^−1^, together with monoanionic diselenolene complexes [AuSePr(=S)]^−1^ and [AuSe*i*Pr(=S)]^−1^ have been investigated and the molecular structures of three of these complexes are shown in Figure 1. The molecular structure of the other complexes is presented in Appendix A. All these complexes surrounded by two disymmetric R-thiazdt or R-thiazds ligands are obtained as trans isomers, in the solid state, with a square planar geometry around the gold atom. Besides the [AuSEt(=C(CN)_2_)]^−1^ complex, all the other monoanionic complexes are not fully planar, as the metallacycles are slightly distorted along the S•••S or Se•••Se axis with dihedral angle in the range of 2 to 6° (see the lateral views of the complexes in Figure 1). The orientation of the chain with alkyl flexible groups, R = Et, Pr, Bu is lateral, while with the bulky isopropyl group R = *i*Pr, the two methyl groups are oriented above and below the plane formed by the complex core.

As these complexes are chalcogen-rich derivatives, in the solid state, various short chalcogen•••chalcogen contacts are observed, shorter than the sum of the van der Waals radii of the involved chalcogen atoms. As a reference, the van der Waals contact distances Se•••Se, S•••Se and S•••S amount to 3.80 Å, 3.70 Å and S•••S 3.60 Å, respectively. The only complex where no short chalcogen•••chalcogen contact was observed is [AuSEt(C=C(CN)_2_)]^−1^. For this complex, hydrogen bonding interactions are established between two neighboring gold complexes involving the nitrogen atoms of one nitrile group and the H of the lateral alkyl chain and between another nitrogen atom of another nitrile group and the H of the counter-ion. In order to exemplify the type of intermolecular chalcogen•••chalcogen short contacts existing in the solid state, the detailed interactions are presented in Figure 2 for the complexes [**P**][AuSePr(=S)] and [**P**][AuSEt(=Se)] and in Appendix A for [**P**][AuSPr(=Se)] and [**P**][AuSBu(=S)]. Concerning the diselenolene complex [**P**][AuSePr(=S)], the monoanionic gold complexes interact from the longitudinal side of the molecule. For instance, some interatomic distances such as Se•••Se are as short as 3.29 Å and Se1•••S1 3.43 Å, which represent a reduction ratio to 86.6% and 92.7%, respectively, relative to the van der Waals contact distance (Se•••Se: 3.80 Å, Se•••S: 3.70 Å and S•••S: 3.60 Å).

As noticed for the bis(diselenolene) complex [**P**][AuSePr(=S)], the bis(dithiolene) [**P**][AuSEt(C=Se)] forms layers along the ab plane with several short chalcogen••• chalcogen interactions, shorter than twice the van der Waals radii. For example, on the side along the long axis of the molecule, two short S•••S contacts are identified, S1•••S1 3.25 Å and S1•••S3 3.39 Å, which correspond to 90.3 and 94.2% of the reduction ratio. Nonetheless, an additional short contact involving the exocyclic selenium atoms is also observed, Se•••Se 3.55 Å (RR of 93.4%).

### 2.3. Cytotoxicity in Ovarian Cells

The anticancer activity of the sixteen complexes under study and the reference drugs Cisplatin and Auranofin was evaluated in the A2780 and OVCAR8 ovarian cancer cells. The Cisplatin-resistant cells OVCAR8 were introduced to ascertain the ability of the complexes to overcome cisplatin resistance, since the majority of ovarian tumors eventually recurs in a drug-resistant form. OVCAR8 were used as a control cell line, selecting the incubation time point of 48 h.

Cells were exposed to increasing concentrations of the complexes (0.01–100 μM) for 24 and/or 48 h, at 37 °C. The IC_50_ values were calculated from dose–response curves using the colorimetric MTT assay. Results show that the differences in the cytotoxic profile were more evident at 24 h incubation, whereas at 48 h incubation most of the complexes displayed similar high cytotoxic activities with IC_50_ values ranging from 0.1–8 µM (A2780) and 0.8–29 µM (OVCAR8). Most complexes presented a similar cytotoxicity profile compared to Auranofin, but higher activity compared to Cisplatin, and, most importantly, were active in the Cisplatin-resistant OVCAR8 cells (Table 2).

Interestingly, in relation to their anticancer activity, and when compared at 48 h incubation, these gold complexes clustered in two groups, i.e., those having Ph_4_P^+^ as a counter-ion and those having Et_4_N^+^ as a counter-ion, being the former more cytotoxic than the later ones. This is even more obvious when comparing the same gold complexes [**N**][AuSEt(C=S)] vs. [**P**][AuSEt(C=S)] (1.30 vs. 0.18), [**N**][AuSeEt(C=Se)] vs. [**P**][AuSeEt(C=Se)] (4.84 vs. 0.64), and [**N**][AuSe*i*Pr(C=S)] vs. [**P**][AuSe*i*Pr(C=S)] (8.27 vs. 0.68). Of note, if some discrepancies can be observed after 24 h incubation, the nature of the exocyclic chalcogen atom do not significantly modify the IC_50_ after 48 h incubation for the A2780 ovarian cancer cells. However, for OVCAR8 ovarian cancer cells, the presence of selenium atoms in the metallacycles seems to diminish the efficiency of the gold complexes.

### 2.4. Toxicity Studies in Zebrafish Embryo

Toxicity is a major challenge for drug development and is the leading cause of drug failure. Consequently, the evaluation of the potential toxicity represents a critical step in the development of new drugs. In this context, the zebrafish embryo represents a notable in vivo vertebrate model for studying the toxic effects of chemical compounds due to the small size, robustness, multiple progenies from a single mating, chemical permeability, easiness of observation and maintenance under laboratory conditions. In addition, the close phylogeny of zebrafish and mammals, with highly conserved signaling pathways, makes this species increasingly used and ideal for performing toxicological studies [52,53,54,55,56]. Therefore, we evaluated the complexes acute toxicity by measuring mortality of 72 h post-fertilization (hpf) zebrafish embryos when incubated with increasing concentrations of the Au(III) complexes for 48 h.

The zebrafish lethality curves for each complex after 48 h of treatment are presented in Appendix A and Figure 3 for selected complexes. Complexes were evaluated at concentrations ranging from 0.1–10 μM. For all the investigated complexes having Et_4_N^+^ as a counter-ion, the survival was affected upon 24 h of treatment for concentrations higher than 1 μM. Contrariwise, for the complexes with Ph_4_P^+^ as counter-ion, no toxicity was evident except for the complex with an exocyclic oxygen atom, [**P**][AuSEt(C=O)]. The nature of the exocyclic atom or group of atoms (=O, =S, or (=C(CN)_2_) seemed to have an impact on the effect of the gold complexes in the zebrafish embryo model. In fact, [**P**][AuSEt(=O)] was more toxic than [**P**][AuSEt(=S)] (=O vs. =S), and [**N**][AuSEt(=C(CN)_2_)] was more toxic than [**N**][AuSEt(=S)] (=C(CN)_2_ vs. =S). Importantly, all the complexes with a Ph_4_P^+^ as a counter-ion of formulae [**P**][AuXR(=Y)], with high antitumor effect on cancer cell lines (Table 2), showed low mortality, an interesting property in preclinical studies for novel therapeutic agents.

### 2.5. Antiplasmodial Activity

Au(III) bis(dithiolene/diselenolene) complexes were initially screened for their activity against the obligatory and clinically silent hepatic stage of the rodent *P. berghei* parasite’s life cycle (Figure 4). Atovaquone (ATQ), a standard antiplasmodial drug, was employed as a positive control, displaying nanomolar-ranged activity. In the negative control, DMSO was incubated at a percentage equivalent to that of the highest compound concentrations tested. All compounds reduced infection at 5 µM. However, in the case of [**P**][AuSEt(=S)], [**P**][AuSEt(=Se)], [**P**][AuSPr(=Se)], [**N**][AuSMe(=S)], [**P**][AuSEt(=O)] and [**N**][AuSEt(=C(CN)_2_)], this was accompanied by some degree of toxicity, as observed by the lower cell confluency values. Eight complexes, [**P**][AuSePr(=S)], [**P**][AuSe*i*Pr(=S)], [**P**][AuSEt(=S)], [**P**][AuSEt(=Se)], [**P**][AuSPr(=Se)], [**P**][AuSeEt(=Se)], [**N**][AuSMe(=S)] and [**P**][AuSBu(=S)] presented the strongest activity and were consequently selected for IC_50_ determination (Table 3).

Complex [**P**][AuSEt(=Se)] was the most active, with an estimated IC_50_ of 415.67 ± 122.73 nM, whereas [**P**][AuSe*i*Pr(=S)] was the least active, with an estimated IC50 of 748.45 ± 70.22 nM. The main difference between these two complexes lies in the nature of the chalcogen within the metallacycle and the exocyclic chalcogen atom. The gold bis(diselenolene) complexes bearing the same exocyclic substituent are less active than the corresponding gold bis(dithiolene) complexes. Among the compounds that presented an interesting liver stage activity, [**P**][AuSEt(=Se)], [**P**][AuSPr(=Se)], [**P**][AuSEt(=S)] and [**P**][AuSBu(=S)] were selected for assessment against the blood stage of *P. falciparum* infection. In this assay, [**P**][AuSEt(=S)] and [**P**][AuSEt(=Se)] displayed the strongest activity against *Pf*NF54 blood stages (Figure 5) and were consequently selected for IC_50_ determination (Table 3).

### 2.6. Antibacterial and Antifungal Activity

The antibacterial activity and antifungal activity of the complexes under study were assessed based on their MIC values against *E. coli*, *S. aureus*, *C. albicans* and *C. glabrata*. Results obtained are presented in Table 4.

All the complexes that demonstrated activity against *S. aureus* have the tetraethylammonium counter-ion Et_4_N^+^. Interestingly, the same complexes with the tetraphenyl phosphonium counter-ion, Ph_4_P^+^, are not active, showing again the influence of the counter-ion in the activity of these complexes. Moreover, the nature of the exocyclic substituent also induces some modulation in the activity of the complexes against *S. aureus* as only those with an exocyclic sulfur atom, =S, show minimum inhibitory concentrations. The two complexes exhibiting the closest MIC values to Auranofin (0.2 µg/mL) are the diselenolene gold complex [**N**][AuSe*i*Pr(=S)] (1.5 ± 0.1 µg/mL) and the dithiolene gold complex [**N**][AuSPr(=S)] (1.5 ± 0.1 µg/mL).

None of the complexes were able to inhibit the growth of *E. coli*. This observation is in line with results obtained previously with gold(III) bis(dithiolene) complexes [46,47].

*C. glabrata* was the most susceptible to the Au(III) complexes, having its growth affected by six of these complexes, namely [**N**][AuSEt(=S)], [**N**][AuSMe(=S)], [**N**][AuSeEt(=S)], [**N**][AuSeEt(=Se)], [**N**][AuSPr(=S)] and [**N**][AuSe*i*Pr(=S)], being the MIC value for last five complexes between 1.6 and 2.9 µg/mL. All these active complexes bear the tetraethylammonium counter-ion (Et_4_N^+^).

The complex [**N**][AuSPr(=S)] was the most active against *C. albicans*, with a MIC value of 3.1 µg/mL, below the 7.9 µg/mL found for Auranofin.

All the complexes with the tetraphenylphosphonium (Ph_4_P^+^) counter-ion were inactive against *S. aureus*, *E. coli*, *C. albicans* and *C. glabrata*.

### 2.7. Cytotoxicity and Anti-HIV Activity of Auranofin, [P][AuSEt(=S)] and [P][AuSBu(=S)]

No significant cytotoxicity was observed in vitro in TZM-bl culture cells in the presence of [**P**][AuSEt(=S)] and [**P**][AuSBu(=S)] (Appendix A). Auranofin and [**P**][AuSEt(=S)] presented a CC_50_ of 3.21 µM and 4.92 µM, respectively. For [**P**][AuSBu(=S)], it was not possible to calculate the CC_50_ (Appendix A). The activity of Auranofin, [**P**][AuSEt(=S)] and [**P**][AuSBu(=S)] was evaluated in TZM-bl cells in a single-round infectivity assays against HIV-1 SG3.1 reference isolate. Similar to Auranofin, none of the compounds presented anti-HIV-1 activity (Figure 6). Of note, Auranofin was considered a candidate anti-HIV reservoir as it targets the memory T lymphocytes which are the viral reservoirs of HIV-1 [57].

### 2.8. Mechanistic Studies

The inhibition of thioredoxin reductase (TrxR), the interaction with DNA and the interaction with human serum albumin were assessed for selected complexes to investigate the effect of the substituent on the nitrogen atom of the thiazoline ring, the nature of the exocyclic atom and the counter-ion.

#### 2.8.1. Inhibition of TrxR by the Gold Complexes

The assessment of TrxR as a possible target for antitumor drugs has attracted wide attention as this enzyme participates in the regulation of redox reactions and other important physiological processes, such as cell proliferation, cell differentiation and cell death [58]. Gold complexes typically behave as strong inhibitors of TrxR, possibly due to the high affinity of gold to its active site selenocysteine. Moreover, gold complexes bearing sulfur-containing ligands display stronger association to TrxR [59,60].

The effect of the selected gold complexes on the activity of TrxR was evaluated by a colorimetric DTNB assay as previously described [47]. Figure 7 shows the effect of the complexes tested in a concentration range of 0.1 nM–2 µM, with Auranofin included in the assay for comparative purposes. As shown in Figure 7 and Appendix A, the inhibitory effect of the complexes differs, although suggesting TrxR as a possible biological target for these complexes. The most active TrxR inhibitors were [**P**][AuSBu(=S)], [**P**][AuSe*i*Pr(=S)] and [**P**][AuSEt(=S)], displaying IC_50_ values of 0.13 ± 0.08 μM, 0.40 ± 0.25 μM and 0.86 ± 0.32 μM, respectively. In contrast, the complex, [**P**][AuSEt(=Se)] with an exocyclic selenium atom was the weaker inhibitor of TrxR, displaying an IC_50_ value of 4.81 ± 1.02 μM. Moreover, it can be noticed that between [**P**][AuS*ei*Pr(=S)] and [**N**][AuSe*i*Pr(=S)], where the only difference is the counter-ion, the weakest effect is with the Et_4_N^+^, indicating that the nature of the counter-ion has a potential effect on the enzyme. The substituent on the nitrogen atom of the thiazoline ring ([**P**][AuSEt(=S)] vs. [**P**][AuSBu(=S)]) and the nature of the exocyclic atom ([**P**][AuSEt(=S)] vs. [**P**][AuSEt(=Se)]) seemed to influence the inhibitory effect on the enzyme. The exocyclic selenium atoms of [**P**][AuSEt(=Se)] seemed to disfavor the intermolecular interaction with the enzyme in relation with the peripheral thione group of [**P**][AuSEt(=S)]. As expected, Auranofin displayed the lowest IC_50_ value, at least one order of magnitude lower. The TrxR inhibitory potential of these complexes is comparable to that of the [Au(R-thiazdt)_2_]^−1^ complexes previously described [47].

TrxRs from different organisms, such as cells, bacteria and *Plasmodium falciparum*, show different chemical mechanisms of reduction, which constitute the basis for the development of specific TrxR inhibitors against bacterial pathogens and parasitic diseases such as malaria. Most protozoan parasites are fast replicating organisms compared with tumor cells and share the same requirements, i.e., rapid DNA synthesis and protection from ROS generated by the host’s immune system. The thioredoxin system can fulfil both tasks and therefore seems to be an essential part of the parasite’s metabolism. On the basis of our results, it is likely that most of the complexes impair the antioxidant system of the parasites [61,62,63,64]. On the other hand, many Gram-positive and Gram-negative bacteria including *S. aureus* and *E. coli*, respectively, possess the Trx-TrxR system to maintain the thiol-redox balance and also to protect bacterial cells from oxidative stress. However, *E. coli* also has the additional glutathione system to maintain the thiol-redox balance. Auranofin inhibits the bacterial TrxR and impair their redox balance, resulting in bacterial cell death. In our study, the activities against Gram-positive and Gram-negative bacteria are consistent with other previous reports indicating a high activity against *S. aureus* (MIC = 0.2 μg/mL) and a significant lack of activity for *E. coli.* (MIC = 35.5 μg/mL) [65].

#### 2.8.2. DNA Electrophoresis

Mechanistic studies suggested that, in contrast to Cisplatin, DNA was not the primary target for gold complexes. In fact, gold has different oxidation states, rich coordination chemistry, and therefore, it is possible to tune their biological activities by subtle changes in the structure of the complexes. Gold(III) complexes have been synthesized to reproduce the main features of Cisplatin, i.e., its planar geometry. Unlike Cisplatin, the main biological targets of gold compounds are still unknown. Several reports mention the DNA-independent mechanism of some gold(III) complexes, while others report that binding to DNA is responsible for the cytotoxic effect [16]. We determined the ability of a set of complexes under study to interact with ΦX174 supercoiled DNA in vitro. As shown in Figure 8 and Appendix A, none of the compounds were able to induce conformational changes in the DNA, following the same pattern observed for Auranofin [46]. Cisplatin was able to alter the electrophoretic mobility of the nicked and supercoiled forms of DNA in a dose-dependent way (Appendix A). These results suggest that the cytotoxic effect of these complexes is not mediated by their interaction with the DNA, suggesting instead a mechanism of action distinct from that of Cisplatin.

#### 2.8.3. Gold-Complexes–HSA Interaction Studies

Understanding the interaction of a drug with serum proteins is an important aspect to be taken into account in the development of new drugs, since this binding may affect its distribution and transport in the human body, as well as influencing its bioavailability, elimination and toxic effects. Fluorescence spectroscopy was used to investigate the interaction between gold complexes [**P**][AuSEt(=S)], [**P**][AuSEt(=Se)], [**P**][AuSe*i*Pr(=S)] and [**P**][AuSBu(=S)] with HSA, taking advantage of the intrinsic fluorescence of the latter. The complexes display a strong quenching of the intensity of the HSA fluorescence as a function of increasing concentration of gold complexes, indicative of a strong interaction between the Au complexes and this protein, without changes in the emission maximum (Figure 9a–d). Upon incubation of two molar excesses of complexes [**P**][AuSEt(=S)], [**P**][AuSEt(=Se)], [**P**][AuSe*i*Pr(=S)] and [**P**][AuSBu(=S)] with HSA, the fluorescence intensity was quenched by 76.3%, 79.4%, 60% and 89.7%, respectively. No intrinsic fluorescence was observed for the gold compounds, in the range 295–550 nm under our experimental conditions and therefore there is no contribution to the HSA fluorescence.

To gather information about the nature of the interaction of the gold complexes with HSA, the Stern–Volmer relationship was applied (Equation (S2)). As illustrated in the insets of Figure 9a–d, the Stern–Volmer plots for the quenching of HSA by the gold complexes showed a good linearity within the tested concentrations, indicating that one type of mechanism, dynamic or static, is involved in this process. The binding constants (K_SV_) deduced from the slope of the linear plot and number of binding sites (n) determined from Equation (S3) are presented in Appendix A. The K_SV_ values followed the pattern [**P**][AuSBu(=S)] ˃ [**P**][AuSEt(=S)] ˃ [**P**][AuSEt(=Se)] ˃ [**P**][AuSe*i*Pr(=S)]. The bimolecular quenching constants (K_q_) estimated from Equation (S2) (Appendix A) are much greater than the scattering collision quenching constant (2.0 × 10^10^ dm^3^ mol^−1^ s^−1^) [66], suggesting that the quenching is mainly due to the formation of ground state complex exclusively and not initiated by the collision/dynamic process.

Overall, the fluorescence spectroscopy indicates that the binding of the gold complexes to the HSA proceeds through a single interaction mode that strongly affects the protein emission, consistent with the formation of a non-fluorescent 1:1 adduct in the ground state for all complexes, except for the complex [**P**][AuSEt(=Se)] which seems to form a non-fluorescent 1:2 adduct. The binding constants determined for these four complexes are higher than that found for Cisplatin (10^2^ M^−1^) and similar to those found for other gold complexes (10^3^−10^4^ M^−1^) [67,68,69,70,71].

## 3. Materials and Methods

### 3.1. General Methods for Chemistry

All reagents and materials from commercial sources were used without further purification. NMR spectra were recorded at room temperature using CDCl_3_ unless otherwise stated (Appendix A). Chemical shifts are reported in ppm and ^1^H NMR spectra were referenced to residual CHCl_3_ (7.26 ppm) and ^13^C NMR spectra were referenced to CHCl_3_ (77.2 ppm). Mass spectra were recorded with an Agilent 6510 instrument for organics compounds and with a Thermo-Fisher Q-Exactive instrument for complexes by the Center Régional de Mesures Physiques de l’Ouest, Rennes. Cyclic voltammograms (CV)s were carried out on a 10^−3^ M solution of complex in CH_2_Cl_2_-[NBu_4_] [PF_6_] 0.1 M. CVs were recorded on a Biologic SP-50 instruments at 0.1 Vs^−1^ on a platinum disk electrode. Potentials were measured versus KCl Saturated Calomel Electrode (SCE). Column chromatography was performed using silica gel Merck 60 (70–260 mesh). The solvents were purified and dried by standard methods. The proligand **2-SEt(=C(CN_2_))**, [72] and complexes [**N**][AuSMe(=S)], [73] [**N**][AuSEt(=S)], [74] [**P**][AuSEt(=O)], [75] [**N**][AuSeEt(=S)] [75] and [**P**][AuSPr(=S)] [76] were synthesized according to the previously reported procedure.

### 3.2. General Synthetic Procedures

#### 3.2.1. Typical Procedure for the Synthesis of the Dithiolene and Diselenolene Proligand 2-XR(=Y):

Under inert atmosphere at 0 °C, lithium diisopropylamide (LDA) was prepared by adding *n*BuLi (2.9 mL, 4.71 mmol, 1.6 M in hexane) to a solution of diisopropylamine (0.7 mL, 4.71 mmol) in 10 mL of anhydrous THF. The LDA solution was added to a solution of **1-R(=Y)** (3.14 mmol) in 40 mL of anhydrous THF at −10 °C. After stirring for 30 min, sulfur for dithiolene or selenium for diselenolene (4.71 mmol) was added to the reaction mixture, and the medium was stirred for an additional 30 min before the addition of LDA (6.28 mmol prepared from 3.9 mL of *n*BuLi, 1.6 M in hexane and 0.9 mL of diisopropylamine in 15 mL of anhydrous THF). The reaction mixture was stirred for 3 h, and sulfur (for dithiolene) or selenium for diselenolene was added (6.28 mmol), followed 30 min later by the addition of 3-bromopropionitrile (2.6 mL, 31.4 mmol). Then, the temperature was allowed to rise to room temperature and the reaction mixture was stirred overnight. Tetrahydrofuran (THF) was evaporated and the residue was extracted with dichloromethane. The organic phase was washed with water and dried over MgSO_4_. The resulting oil was purified by column chromatography on silica gel using CH_2_Cl_2_ as eluent.

**2-SBu(=S),** brown oil in 59% yield; Rf = 0.45 (SiO_2_, CH_2_Cl_2_); ^1^H NMR (300 MHz) δ 1.03 (t, 3H, CH_3_, *^3^J* = 7.3 Hz), 1.45 (m, 2H, CH_2_), 1.75 (m, 2H, CH_2_), 2.75 (t, 4H, CH_2_, *^3^J* = 7.0 Hz), 3.14 (t, 2H, CH_2_, *^3^J* = 7.0 Hz), 3.20 (t, 2H, CH_2_, *^3^J* = 7.0 Hz), 4.39 (t, 2H, CH_2_, *^3^J* = 7.3 Hz); ^13^C NMR (75 MHz) δ 13.7 (CH_3_), 18.7 (CH_2_-CN), 19.9 (CH_2_-CH_3_), 29.8 (CH_2_-CH_2_-N), 31.7 (S-CH_2_), 32.4 (S-CH_2_), 49.5 (CH_2_-N), 117.6 (CN), 117.8 (CN), 125.7 (C=C), 136.1 (C=C), 187.3 (C=S); HRMS (ESI) calcd for C_13_H_17_N_3_S_4_^+•^: calcd 343.0305. Found: 343.0335.

**2-SePr(=S),** beige powder in 51% yield; mp = 142 °C. R_f_ = 0.54 (SiO_2_, CH_2_Cl_2_/Et_2_O, 9.8/0.2). ^1^H NMR (300 MHz) δ 0.99 (t, 3H, CH_3_, *^3^J* = 7.5 Hz), 1.76 (sextuplet, 2H, CH_2_, *^3^J* = 7.5 Hz), 2.87 (t, 4H, CH_2,_
*^3^J* = 6.9 Hz), 3.11 (t, 4H, CH_2,_
*^3^J* = 6.9 Hz), 4.35 (t, 2H, CH_2_, *^3^J* = 7.5 Hz). ^13^C NMR (75 MHz) δ (10.9 (CH_3_), 19.3 (Se-CH_2_-CH_2_-CN), 21.5 (CH_3_-CH_2_-CH_2_-N), 24.2 (Se-CH_2_-CH_2_-CN), 52.5 (CH_3_-CH_2_-CH_2_-N), 117.8 (CN), 131.1 (C=C), 188.9 (C=S). HRMS (ASAP) calcd for C_12_H_16_N_3_S_2_Se_2_^+•^: 425.91161, found: 425.9111. Anal. calcd for C_12_H_15_N_3_S_2_Se_2_: C, 34.04; H, 3.57; N, 9.92; S, 15.15. Found: C, 33.90; H, 3.45; N, 9.23; S, 14.87.

**2-Se*i*Pr(=S),** orange oil in 50 % yield; R*_f_* = 0.43 (SiO_2_, CH_2_Cl_2_/Et_2_O, 9.8/0.2). Two rotamers can be observed on the NMR spectra. ^1^H NMR (300 MHz) δ *main rotamer* (56.2%): 1.80 (d, 6H, CH_3_, *^3^J* = 7.0 Hz), 3.02 (t, 8H, CH_2_, *^3^J* = 7.1 Hz), 5.27 (sept, 1H, CH, *^3^J* = 7.0 Hz); *other rotamer* (43.8%): 1.64 (d, 6H, 2CH_3_, *^3^J* = 7.2 Hz), 2.97 (t, 4H, CH_2_, *^3^J* = 7.1 Hz), 3.08 (t, 4H, CH_2_, *^3^J* = 7.1 Hz), 5.96 (sept, 1H, CH, *^3^J* = 7.2 Hz). ^13^C NMR (75 MHz) δ *main rotamer:* 18.0 (N-CH(CH_3_)_2_), 20.2 (Se-CH_2_-CH_2_-CN), 25.2 (Se-CH_2_-CH_2_-CN), 53.7 (CH), 118.9 (CN), 126.1 (=C), 128.4 (=C), 189.3 (C=S); *other rotamer:* 21.9 (N-CH(CH_3_)_2_), 21.2 (Se-CH_2_-CH_2_-CN), 27.32 (Se-CH_2_-CH_2_-CN), 58.6 (CH), 119.0 (CN), 126.3 (=C), 131.7 (=C), 187.7 (C=S). HRMS (ASAP) calcd for C_12_H_15_N_3_S_2_Se_2_^+•^: 425.91161. Found: 425.9115.

#### 3.2.2. Procedure for the Synthesis of Monoanionic Gold Bis(Dithiolene) and Bis Diselenolene Complexes:

To a dry two-neck flask containing the **2-XR(=Y)** (0.8 mmol), 7.5 mL of 1M solution of sodium methoxide in MeOH was added under nitrogen atmosphere at room temperature. After stirring for 30 min, a solution of potassium tetrachloroaurate (III) hydrate (KAuCl_4_.H_2_O) (0.4 mmol) in 7 mL of anhydrous methanol was added to the reaction mixture followed, 5 h later, by the addition of tetraphenylphosphonium chloride Ph_4_PCl or Et_4_NBr (0.4 mmol) in 7 mL of anhydrous methanol. The reaction was stirred overnight at room temperature under argon atmosphere. The dark brown solution was filtered over a vacuum flask. The resulting solid was washed with MeOH and recrystallized from acetonitrile.

**[P][AuSePr(=S)]**, black crystals, in 61% yield; mp = 172 °C; ^1^H NMR (300 MHz) δ 0.97 (t, 6H, CH_3_, *^3^J* = 7.5 Hz), 1.77 (m, 4H, CH_2_, *^3^J* = 7.6 Hz), 4.05 (t, 4H, CH_2_, *^3^J* = 7.6 Hz), 7.61 (m, 8H, CH_Ar_), 7.74 (m, 8H, CH_Ar_), 7.89 (m, 4H, CH_Ar_); ^13^C NMR (75 MHz) δ (ppm): 11.0 (N-CH_2_-CH_2_-CH_3_), 21.0 (N-CH_2_-CH_2_-CH_3_), 50.2 (N-CH_2_-CH_2_-CH_3_), 104.7 (C=C), 116.5 (C_Ar_), 117.7 (C_Ar_), 128.7 (C=C), 130.5 (C_Ar_), 134.0 (C_Ar_), 135.6 (C_Ar_), 192.6 (C=S); UV-vis-NIR (CH_2_Cl_2_) λ (nm), ε (M^−1^.cm^−1^): 362, 32455; 314, 27964; 260, 26167; 224, 44491. HRMS (ESI) calcd for [2C^+^, A^−^]^+^ [C_60_H_54_N_2_P_2_S_4_Se_4_Au]^+^: 1508.89661, found: 1508.8991. Anal. calcd for C_36_H_34_AuN_2_PS_4_Se_4_: C, 37.06; H,2.94; N, 2.40; S, 10.99. Found: C, 36.59; H, 3.17; N, 2.31; S, 10.69.

**[P][AuSe*i*Pr(=S)]**, black-green crystals, in 48% yield; mp = 216 °C; two rotamers can be observed on the ^1^H NMR spectrum realized at 243 K the main one at 87% and the other at 13% ^1^H NMR (300 MHz) δ 1.57 (broad signal, 12H, CH_3_, 87%), 1.60 (broad signal, 12H, CH_3_, 13%), 4.56 (broad signal, 2H, CH, 13%), 5.83 (broad signal, 2H, CH, 87%), 7.60 (m, 8H, CH_Ar_), 7.77 (m, 8H, 8CH_Ar_), 7.93 (m, 4H, CH_Ar_). UV-vis-NIR (CH_2_Cl_2_) λ (nm), ε (M^−1^.cm^−1^): 364, 28982; 318, 32095; 260, 29341; 224, 54311; HRMS (ESI) calcd for calcd for [2C^+^, A^−^]^+^ [C_60_H_54_N_2_P_2_S_4_Se_4_Au]^+^: 1508.89661, found: 1508.8976. Anal. calcd for C_36_H_34_AuN_2_PS_4_Se_4_: C, 37.06; H,2.94; N, 2.40; S, 10.99. Found: C, 36.64; H, 3.12; N, 2.38; S, 10.98.

**[P][AuSeEt(=Se)]**, black-green crystals, in 45% yield; mp = 216 °C; ^1^H NMR (300 MHz) δ 1.33 (t, 6H, CH_3_, *^3^J* = 7.2 Hz), 4.28 (q, 4H, CH_2_, *^3^J* = 7.2 Hz), 7.60 (m, 8H, CH_Ar_), 7.75 (m, 8H, CH_Ar_), 7.85 (m, 4H, CH_Ar_). HRMS (ESI) calcd for [2C^+^, A^−^]^+^ [C_58_H_50_AuN_2_P_2_S_2_^80^Se_6_]^+^: 1576.7542. Found: 1576.7573. Anal. calcd for C_34_H_30_AuN_2_PS_2_Se_6._ CH_2_Cl_2_: C, 31.91; H, 2.45; N, 2.13; S, 4.87. Found: C, 32.08; H, 2.46; N, 2.09; S, 5.21.

**[P][AuSEt(=Se)]**, black-green crystals, in 20% yield; mp = 214 °C; ^1^H NMR (300 MHz) δ 1.30 (t, 6H, CH_3_, *^3^J* = 7.2 Hz), 4.20 (q, 4H, CH_2_, *^3^J* = 7.2 Hz), 7.62 (m, 8H, CH_Ar_), 7.78 (m, 8H, CH_Ar_), 7.91 (m, 4H, CH_Ar_); HRMS (ESI) calcd for [2C^+^, A^−^]^+^ [C_58_H_50_AuN_2_P_2_S_6_^80^Se_2_]^+^: 1384.9764. Found: 1384.9776; Anal. calcd for C_34_H_30_AuN_2_PS_6_Se_2_: C, 39.08; H,2.89; N, 2.68. Found: C, 38.77; H, 2.44; N, 2.70.

**[N][AuSPr(=S)]**, black-green crystals, in 47% yield; mp = 231 °C; ^1^H NMR (300 MHz) δ 0.98 (m, 6H, CH_3_), 1.32 (m, 12H, CH_2_), 1.81 (m, 4H, CH_2_), 3.25 (q, 8H, CH_2_, ^3^*J* = 7.3 Hz), 4.06 (m, 4H, CH_2_); HRMS (ESI) calcd for [2C^+^, A^−^] [C_28_H_54_AuN_4_S_8_]^+^: 899.17743. Found: 899.1775; Anal. calcd for C_20_H_34_AuN_3_S_8:_ C, 31.20; H,4.45; N, 5.46. Found: C, 30.87; H, 4.44; N, 5.96.

**[P][AuSPr(=Se)]**, black-green crystals, in 47% yield; mp = 228 °C; ^1^H NMR (300 MHz) δ 0.96 (m, 6H, CH_3_), 1.79 (m, 4H, CH_2_), 4.09 (m, 4H, CH_2_), 7.61 (m, 8H, CH_Ar_), 7.76 (m, 8H, CH_Ar_), 7.89 (m, 4H, CH_Ar_); HRMS (ESI) calcd for [2C+A^−^] [C_12_H_14_AuN_2_S_6_^80^Se_2_]^−^: 734.4828. Found: 734.7488. Anal. calcd for C_36_H_34_AuN_2_PS_6_Se_2:_ C, 40.30; H,3.19; N, 2.61. Found: C, 40.24; H, 3.44; N, 2.96.

**[P][AuSBu(=S)],** dark crystals, in 29% yield. mp 225 °C; ^1^H NMR (300 MHz, (CD_3_CN) δ 0.95 (t, 6H, CH_3_, ^3^J = 7.3 Hz), 1.39 (sext, 4H, CH_2_, ^3^J = 7.3 Hz), 1.70 (m, 4H, CH_2_), 4.06 (t, 4H, CH_2_, *^3^J* = 7.3 Hz), 7.69 (m, 16H, Ar), 7.91 (m, 4H, Ar); ^13^C NMR (75 MHz, (CD_3_)_2_SO) δ 13.5 (CH_3_), 19.5 (CH_2_-CH_3_), 29.1 (CH_2_-CH_2_-CH_3_), 46.9 (CH_2_-N), 110.2 (C = C), 117.1 (C_Ar_), 118.3 (C_Ar_), 130.4 (C_Ar_), 134.6 (C_Ar_), 135.4 (C = C), 190.7 (C = S); Anal. calcd for C_38_H_38_AuN_2_PS_8_: C, 45.32; H, 3.80; N, 2.78; S, 25.47. Found: C, 45.47; H, 3.92; N, 2.72; S 24.97.

**[N][AuSEt(=C(CN)_2_)],** 47% yield as dark crystals; Mp > 250 °C; ^1^H NMR (300 MHz, (CD_3_)_2_SO) δ 1.18 (m, 12H, CH_3_), 1.31 (t, 6H, CH_3_, ^3^*J* = 7.1 Hz), 3.21 (q, 8H, CH_2_, ^3^*J* = 7.3 Hz), 4.07 (q, 4H, CH_2_, ^3^*J* = 7.1 Hz); ^13^C NMR (75 MHz, (CD_3_)_2_SO) δ 7.6 (N-(CH_2_-CH_3_)_4_), 15.0 (N-CH_2_-CH_3_), 44.6 (N-CH_2_), 51.9 (N-(CH_2_-CH_3_)_4_) 111.0 (C-CN), 117.3 (C = C), 134.3 (CN), 174.3 (C = C(CN)_2_). HRMS (ESI) calcd for [2C^+^, A^−^]^+^ [C_32_H_50_N_8_S_6_Au]^+^: 935.21428. Found: 935.2143. Anal. calcd for C_24_H_30_AuN_7_S_6_. CH_2_Cl_2_: C, 33.71; H 3.62; N, 11.01; S, 21.59. Found: C, 33.24; H, 3.58; N, N, 10.76; S, 21.85.

**[P][AuSEt(=C(CN)_2_)],** 40% yield as dark crystals; Mp > 250 °C; ^1^H NMR (300 MHz, CD_3_CN) δ 1.34 (t, 6H, CH_3_, ^3^*J* = 7.3 Hz), 4.10 (q, 4H, CH_3_, ^3^*J* = 7.3 Hz), 7.70 (m, 16H, Ar), 7.91 (m, 4H, Ar); ^13^C NMR (75 MHz, (CD_3_)_2_SO) δ 14.9 (CH_3_), 44.3 (CH_2_), 112.1 (C-CN), 117.0 (C = C), 117.5 (C = C), 118.2 (CN), 130.4 (C_Ar_), 134.4 (C_Ar_), 136.1 (C_Ar_), 174.7 (C = C(CN)_2_); HRMS (ESI) calcd for [2C^+^, A^−^]^+^ [C_64_H_50_P_2_S_6_Au]^+^: 1353.15566. Found: 1353.1551. Anal. calcd for C_40_H_30_AuN_6_PS_6_ + 2CH_2_Cl_2_: C, 42.58; H 2.89; N, 7.09. Found: C, 42.44; H, 2.86; N, 7.10.

### 3.3. Crystallography

Data collections were performed on an APEXII Bruker-AXS diffractometer equipped with a CCD camera for [**P**][AuSePr(=S)], [**P**][AuSeiPr(=S)], [**P**][AuSEt(=Se)], [**P**][AuSPr(=Se)], [**P**][AuSBu(=S)], on a D8 VENTURE Bruker AXS diffractometer for [**P**][AuSEt(=C(CN)_2_)]. Apart from [**P**][AuSEt(=C(CN)_2_)], the structures were solved either by direct methods using the *SIR97* program [77] and then refined with full-matrix least-square methods based on F^2^ (*SHELXL-97*) [78] with the aid of the *WINGX* [79] program. For [**P**][AuSEt(=C(CN)_2_)], the structure was solved by direct methods using SIR97 program [77], and then refined with full-matrix least-squares methods based on F^2^ (SHELXL program [80]). All non-hydrogen atoms were refined with anisotropic atomic displacement parameters. H atoms were finally included in their calculated positions. Details of the final refinements are summarized in Appendix A.

### 3.4. Biological Studies

#### 3.4.1. Cytotoxic Activity

The cytotoxic activity of the complexes was evaluated in the A2780 (cisplatin sensitive) (Sigma-Aldrich) and OVCAR8 (cisplatin resistant) (ATCC) ovarian cancer cells. Cells were grown in RPMI-1640 medium (Gibco, Thermo Fisher) supplemented with 10% fetal bovine serum (FBS) and were maintained at 37 °C in a humidified atmosphere with 5% CO_2_. The complexes were initially dissolved in DMSO and then in culture medium to prepare serial dilutions in the range 10^−9^–10^−4^ M. After 24 and/or 48 h incubation the cellular viability was measured by the colorimetric MTT assay, as previously described [46,47].

The in vitro cytotoxicity of auranofin, [**P**][AuSBu(=S)] and [**P**][AuSEt(=S)] was evaluated in TZM-bl cells using alamarBlue cell viability reagent (Life Technologies, Carlsbad, CA, USA) [81]. TZM-bl cells (AIDS Research and Reference Reagent Program, National Institutes of Health, Bethesda, MD, USA) were cultured in complete growth medium that consists of Dulbecco’s minimal essential medium (DMEM) supplemented with 10% fetal bovine serum (FBS), 100 U/ml of penicillin-streptomycin (Gibco/Invitrogen, Waltham, MA, USA), 1 mM of sodium pyruvate (Gibco/Invitrogen, Carlsbad, CA, USA), 2 mM of L-glutamine (Gibco/Invitrogen, USA) and 1 mM of non-essential amino acids (Gibco/Invitrogen, USA). Cells were cultured in the presence and absence of serial-fold dilutions of the compounds. At least two independent experiments were performed for each cytotoxicity analysis. Each dilution of each compound was performed in triplicate wells. For each assay we had medium controls (only growth medium), cell controls (cells without test compound) and cytotoxicity controls (a compound that kill cells). The cytotoxicity of each compound was expressed by the 50% cytotoxic concentration (CC_50_), which is the concentration of compound causing 50% decrease in cellular viability.

#### 3.4.2. Toxicological Assessment in Zebrafish Embryo

To test the toxicological effect of the gold (III) complexes, zebrafish (*Danio rerio*) embryos at developmental stage of 72 h post-fertilization (hpf) were exposed to different concentrations of the target complexes and mortality was assessed for 48 h of treatment. Thus, the incubation of a total of 30 embryos per compound concentration was carried in 6-well plates (5 embryos per well) in zebrafish embryo medium (14 mM NaCl, 0.5 mM KCl, 0.02 mM Na_2_HPO_4_, 0.04 mM KH_2_PO_4_, 1.36 mM CaCl_2_, 2.13 mM MgSO_4_, 4.34 mM NaHCO_3_, prepared in 100 mL dH_2_O). Concentrations used for each compound varied between 0.1 and 10 μM, as described below. The incubation medium was daily renewed. Control DMSO concentration was used as the highest concentration of each gold (III) compound.

#### 3.4.3. In Vitro Activity of Gold Complexes against the Hepatic Stage of *P. berghei* Infection

The in vitro liver stage activity of gold complexes was assessed against *P. berghei*-infected Huh7 cells, as previously described [47,82]. Briefly, Huh7 cells were cultured in RPMI-1640 medium supplemented with 10% (v/v) fetal bovine serum, 1% (v/v) penicillin/streptomycin, 1% (v/v) glutamine, 1% (v/v) non-essential amino acids and 10 mM HEPES. On the day prior to infection, Huh7 cells were seeded in 96-well plates at a density of approximately 3.1 × 10^4^ cell/cm^2^ and incubated at 37 °C, 5% CO_2_. Stock solutions of the complexes were initially prepared in DMSO and serially diluted in infection medium, i.e., culture medium supplemented with gentamicin (50 µg/mL) and fungizone (0.8 µg/mL), to obtain the test concentrations. As a control, DMSO was diluted in infection medium to a percentage that equals that of the highest compound concentrations tested. These dilutions were added to the cells in triplicates and incubated for 1 h at 37 °C, 5% CO_2_, after which 1 × 10^4^ luciferase-expressing *P. berghei* sporozoites were added to each well. Plates were centrifuged at 1800× *g* for 5 min at room temperature and incubated for the duration of the assay. At 46 h post-infection (hpi), the impact of each compound concentration on cell viability was assessed by the Alamar Blue (Invitrogen, Buckinghamshire, UK) assay, according to the manufacturer’s recommendations. Infection was next assessed by bioluminescence, employing the Firefly Luciferase Assay Kit 2.0 (Biotium, Fremont, CA, USA), according to the manufacturer’s recommendations. For IC50 determination, nonlinear regression analysis was employed to fit the normalized results of the dose–response curves, using GraphPad Prism 8 (GraphPad software, La Jolla, CA, USA).

#### 3.4.4. In Vitro Activity of Gold Complexes against the Blood Stage of *P. falciparum* Infection

Ring-stage synchronized cultures of *Pf*NF54 at 2.5% hematocrit and at approximately 1% parasitemia were incubated with gold complexes or DMSO (vehicle control) in 96 well-plates, for 48 h, at 37 °C in a 5% CO_2_ and 5% O_2_ atmosphere. Stock solutions of Chloroquine (positive control), [**P**][AuSEt(=S)] and [**P**][AuSEt(=Se)] were prepared in DMSO. Working solutions were prepared from the stock solutions in complete malaria culture medium (CMCM), which consists of RPMI 1640 supplemented with 25 mM HEPES, 2.4 mM L-glutamine, 50 μg/mL gentamicin, 0.5% w/v Albumax, 11 mM glucose, 1.47 mM hypoxanthine and 37.3 mM NaHCO_3_. For each measurement, 5 µL of the culture (approximately 800,000 cells) was stained with the DNA-specific dye SYBR green I at 1×. After 20 min of incubation, in the dark, the stained sample was analyzed by flow cytometry. For each flow cytometric measurement, approximately 100,000 events were analyzed. All samples were analyzed in triplicate and two independent experiments were performed.

#### 3.4.5. Bacterial and Fungal Strains

The bacterial strains *Escherichia coli* ATCC25922 and *Staphylococcus aureus* Newman were used in this work and maintained in Lennox Broth (LB) solid medium, composed of 10 g/L tryptone, 5 g/L yeast extract, 5 g/L NaCl and 20 g/L agar. The fungal strains *Candida albicans* SC5134 and *C. glabrata* CBS138 were used in this work and maintained in YPD solid medium, composed of 20 g/L glucose, 20 g/L peptone, 10 g/L yeast extract and 15 g/L agar. The strains were isolated from human infections [83,84,85,86].

#### 3.4.6. Antimicrobial Activity

The antimicrobial activity of the complexes towards the bacterial and fungal strains was assessed by the determination of the Minimal Inhibitory Concentration (MIC) of each complex towards the indicated microbial strains, based on standard methods, and as previously described [47,87,88]. Briefly, stock solutions of the complexes were prepared in 100% DMSO at final concentrations of 10 mg/mL (complexes [**N**][AuSPr(=S)], [**P**][AuSePr(=S)], [**N**][AuSe*i*Pr(=S)] and [**N**][AuSMe(=S)]), 5 mg/mL (complexes [**P**][AuSEt(=S)], [**P**][AuSEt(=Se)], [**P**][AuSPr(=Se)], [**N**][AuSeEt(=S)]Y1, [**P**][AuSeEt(=Se)], [**P**][AuSEt(=O)], [**N**][AuSEt(=S)], [**N**][AuSEt(=C(CN)_2_)], [**P**][AuSBu(=S)] and Auranofin) or 2.5 mg/mL (complexes [**P**][AuSe*i*Pr(=S)] and [**N**][AuSeEt(=Se)]), depending on the solubility of each complex. Serial 1:2 dilutions of stock solutions were prepared for each complex in Mueller–Hinton (MH) broth (Fluka Analytical) or RPMI-1640 medium (Sigma) supplemented with 20 g/L glucose and buffered with 0.165 M morpholinepropanesulphonic acid (RPMIG) to pH 7.0. Solutions prepared with MH were used for testing bacteria, while those prepared with RPMIG were used for testing fungi. For antibacterial activity assays, the final concentrations ranged from 250 to 0.24 μg/mL (when stock solution final concentration was 10 mg/mL), 125 μg/mL to 0.12 μg/mL (when stock solution final concentration was 5 mg/mL) and 62.5 to 0.06 μg/mL (when stock solution final concentration was 2.5 mg/mL). For antifungal assays, the final concentrations ranged from 125 μg/mL to 0.12 μg/mL (when stock solution final concentration was 10 mg/mL), 62.5 to 0.06 μg/mL (when stock solution final concentration was 5 mg/mL) and 31.25 to 0.03 μg/mL (when stock solution final concentration was 2.5 mg/mL). Then, 100 μL aliquots of adequately diluted bacterial suspensions of *S. aureus* Newman or *E. coli* ATCC 25922 were mixed with the MH serially diluted complexes aliquots to obtain 5 × 10^5^ CFU/mL. In the case of fungi, cultures were diluted to obtain a final optical density of 0.025, measured at 530 nm (OD_530_). Bacterial suspensions were prepared from cultures grown for 5 h in MH broth at 37 °C and 250 rev·min^−1^ and adequately diluted with fresh MH medium, while fungal suspensions were prepared from overnight cultures grown in YPD broth at 30 °C and 250 rev.min^−1^ and diluted in buffered RPMIG. After 22 h of incubation at 37 °C (bacteria) or 24 h at 35 °C (fungi), the wells’ content was resuspended by pipetting and the OD_640_ (bacteria) or OD_530_ (fungi) were measured in a SPECTROstar Nano microplate reader (BMG Labtech).

At least three independent experiments were performed in duplicate for each complex under study. MIC values were estimated after fitting the OD mean values measured for bacteria and fungi after 24 h of incubation, using a modified Gompertz equation and the GraphPad Prism software (version 6.07) [89]. Positive (no complex) and negative controls (no inoculum) were performed for each experiment. The effect of 5% (V/V) DMSO on microbial growth was also assessed.

#### 3.4.7. Anti-HIV Assays

The HIV-1 SG3.1 subtype B strain was obtained by transfection of HEK293T cells with pSG3.1 plasmid using jetPrime transfection reagent (Polyplus-tranfection SA, Illkirch, France) according to the manufacturer’s instructions. The 50% tissue culture infectious dose (TCID_50_) of the virus was determined in a single-round viral infectivity assay using a luciferase reporter gene assay [90] in TZM-bl cells and calculated using the statistical method of Reed and Muench [91].

The antiviral activity of Auranofin, [**P**][AuSEt(=S)] and [**P**][AuSBu(=S)] compounds was determined in a single-round viral infectivity assay using TZM-bl reporter cells, as previously described [90]. Briefly, TZM-bl cells were infected with 200 TCID_50_ of SG3.1 in the presence of serial-fold dilutions of the compounds in growth medium, supplemented with DEAE-dextran. After 48 h of infection, luciferase expression was quantified with Pierce Firefly Luc One-Step Glow Assay Kit (ThermoFisher Scientific, Rockford, IL, USA) according to the manufacturer’s instructions.

At least two independent experiments were performed for each antiviral activity analysis. The assay was set up in triplicate wells. Virus controls and cell controls were used.

### 3.5. Mechanistic Studies for Relevant Complexes

#### 3.5.1. DNA

The potential interaction of the complexes with DNA was evaluated through the assessment of the electrophoretic mobility of supercoiled ϕX174 DNA, using a previously described method [46]. To that end, a mixture was prepared containing 200 ng of supercoiled ϕX174 DNA (Promega) in 10 mM of phosphate buffer (pH 7.2) and increasing concentrations of Cisplatin, as a positive control, or the metal complexes (in a total volume of 20 μL). The mixture was incubated for 24 h at 37 °C, in the dark. As controls, samples having non-incubated plasmid and plasmid incubated with DMSO were also prepared. Then, all samples were prepared for electrophoresis through the addition of 2 μL of 10× DNA loading buffer (Applichem) and loading on an 0.8% agarose gel in TBE buffer (Thermo Fisher Scientific, Waltham, MA, USA). The gel was run at 90 V for approximately 3 h, stained using a 3× GelRed^®^ (Biotium) solution in H_2_O, and the bands visualized using an AlphaImagerEP (Alpha Innotech) under UV light.

#### 3.5.2. Thioredoxin (TrxR) Inhibition Study

The thioredoxin (TrxR) inhibition assays were performed using a commercial kit from Sigma-Aldrich, introducing minor modifications for a 96-well plate format. The assay is based on the reduction of DTNB (5,5′ dithiobis(2-nitrobenzoic acid)) into TNB (5-thio-2-nitrobenzoic acid) with the concomitant oxidation of NADPH, in a reaction catalyzed by TrxR [92]. The complexes were dissolved in DMSO to prepare serial concentrations in the range of 0.1 nM–2 µM. The reaction mixture (200 µL total volume) contained 11 μL assay buffer (phosphate buffer, pH 7.0 and 50 mM EDTA), 180 µL working buffer (phosphate buffer and 0.25 mM NADPH), 1 μL complexes’ solutions, 2 μL enzyme solution and 6 µL DTNB (0.1 M in DMSO). A blank sample (no enzyme) and a positive control (no complexes) were included in the assays. The formation of TNB was monitored at 412 nm with a plate spectrophotometer (Power Wave Xs, Bio-Tek).

#### 3.5.3. HSA-Binding Experiments

##### Sample Preparation for Spectrofluorometric Experiments

Stock solutions of HSA were prepared by dissolving HSA in 10 mM Hepes buffer, pH 7.4. The protein concentration was determined spectrophotometrically using the molar absorption coefficient of 36,500 M^−1^ cm^−1^ at 280 nm [93]. The complexes [**P**][AuSEt(=S)], [**P**][AuSEt(=Se)], [**P**][AuSe*i*Pr(=S)] and [**P**][AuSBu(=S)] were first prepared at 1 mM concentration in DMSO due to the limited solubility of the complexes in aqueous media. A series of complex–protein solutions were prepared by adding different concentrations of gold complexes solutions to the protein solution previously prepared. After preparation of the batch complex–protein solutions, the final concentration of DMSO was 1%. For fluorescence acquisition, the final HSA concentration was 2.5 mM and the gold concentrations were 0, 0.6, 1.3, 1.9, 2.5, 3.1, 3.8, 4.4 and 5 mM. The mixtures were stirred to ensure the formation of a homogeneous solution and then kept in an incubator at 37 °C for 24 h in the dark to stabilize and enhance the interaction time. The reference solutions without protein were prepared following the procedures described above.

##### Fluorescence Spectroscopic Measurements

Steady-state fluorescence measurements were performed with a Fluorolog Model-3.22 spectrofluorimeter from Horiba Jobin Yvon at 25 °C. All the experiments were performed in Hellma^®^ semi-micro fluorescence cuvettes (Suprasil^®^ quartz, path-length 10 × 4 mm, chamber volume 1.4 μL) with the 10 mm path length for the excitation of the sample. The excitation and emission bandwidths were fixed at 4.0 nm, and the excitation wavelength was 295 nm to excite selectively the tryptophan 214 residue, the emission spectra were recorded from 305 to 550 nm. Buffer solutions of gold complexes in corresponding concentrations were used as reference for the measured fluorescence spectra of protein–complex mixtures. The fluorescence intensities were corrected for the absorption of the exciting light and reabsorption of the emitted light to decrease the inner filter effect [94,95] using UV–visible absorption data recorded for each sample on a Jasco V-660 spectrophotometer in the range of 260 to 800 nm with 1 mm path quartz cells. More details and all the equations can be found in the SI.

## 4. Conclusions

In the present study, sixteen structurally related monoanionic gold(III) bis(dithiolene/diselenolene) complexes were synthesized and characterized by standard analytical techniques. These gold(III) complexes with dithiolene ligands are redox-active molecules which are known as excellent precursors of molecular materials exhibiting conducting properties. Within this study we explored the prospective pharmacological use of these gold complexes, namely as anticancer, antiplasmodial, antibacterial, antifungal and anti-HIV agents. These complexes feature square planar geometries, such as Cisplatin. Based on the molecular structure, a similar mechanism of action, e.g., interaction with DNA, would be expected for these complexes. Reports on the biological activity of similar gold(III) complexes are relatively scarce in the literature. Moreover, these complexes are monoanionic, which in theory do not favor their uptake into cells due to the negatively charged plasma membrane. In this regard, studies on the mechanism of uptake of prospective metal drugs are relatively rare and, similar to Cisplatin, the uptake usually occurs by more than one mechanism.

Cisplatin and Auranofin were included in this study as positive controls. The mode of action of Cisplatin relies on the binding to nuclear DNA, interfering with normal transcription and/or DNA replication mechanisms. Several targets have been proposed to be involved in the action of Cisplatin. In fact, before reaching the target DNA, many other cellular biomolecules such as sulfur-containing glutathione and metallothionein can be targeted and deactivated. The main mechanism of action of Auranofin is through the inhibition of redox enzymes such as TrxR. The overexpression of TrxR is associated with aggressive tumor progression and poor survival in patients with several types of cancers. The thiol group in Auranofin structure has a high affinity to bind thiol and selenol groups of proteins, thus forming stable and irreversible adducts. Redox enzymes such as TrxR are essential to many cellular processes, particularly in maintaining the intracellular levels of reactive oxygen species (ROS) to prevent the resulting DNA damage. An alternative mechanism of action of Auranofin is through the inhibition of the ubiquitin–proteasome system in cancer cells, which are involved in cell cycle regulation, protein degradation, gene expression and DNA repair.

Most of the gold complexes herein exhibited strong anticancer activities similar to Auranofin. However, regarding the antiplasmodial, antibacterial and antifungal activities, most of the complexes displayed selective activities, i.e., those that present outstanding cytotoxic activities also display high antiplasmodial activities, which contrasts with their antibacterial and antifungal properties. Complexes´ features, in particular the counter-ion and the nature of the exocyclic atom or group of atoms, seemed to be relevant for their biological activities. Similar to Auranofin, the complexes failed to interact with ΦX174 supercoiled DNA, indicating that DNA is not their cellular target. In contrast, complexes were able to interact with TrxR and inhibit its activity, although to a lesser extent than Auranofin. HSA could be a possible vehicle of transport for these gold complexes due to their strong affinities for the protein. This result constitutes an interesting finding, as the therapeutic value of a prospective drug is dependent on its availability at the target site. The mode of binding to albumin is central to understanding the pharmacokinetic profile and has a major influence on the therapeutic efficacy.

In this contribution, the main findings are the influence of the counter-ions on the biological properties of these complexes: (i) the complexes with a phosphonium ([P]) counter-ion in general favor the anticancer and antiplasmodial activity; (ii) the complexes with an ammonium ([N]) counter-ion favor the antibacterial/fungi activities; and (iii) the complexes with [P] counter-ion exhibited low toxicity in the zebrafish embryo model. Another feature is the nature of the exocyclic chalcogen atom, e.g., the exocyclic sulfur replacement by = C(CN)_2_ increases the cytotoxic activity and the steric hindrance generated by the side chain, from Pr to *i*Pr, decreases the anticancer activity. Although this study uses a relatively modest set of compounds for a SARs approach revealing relationships between structural properties and biological activities, it represents the basis for the development of potent pharmaceutical agents.

In summary, our results show that these complexes are potentially valuable drug candidates and suitable for further pharmacological testing, in particular those that show very low toxicity in zebrafish embryos. Similar to cancer cells, parasites need to maintain cellular redox balance and rely on redox enzymes such as TrxR to control the levels of ROS in the cytosol and mitochondria. Considering the similarities between the basic biological aspects of cancer and parasites and the fact that some compounds developed for cancer treatment could also be able to target parasitic enzymes such as Auranofin, our gold complexes could be explored for the benefit of both conditions, placing these compounds on the cutting edge of new research approaches in medicinal chemistry.

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
