# Peer review of "Broad Spectrum Functional Activity of Structurally Related Monoanionic Au(III) Bis(Dithiolene) Complexes"

_ijms, 2022, doi:10.3390/ijms23137146_

Round 1

Reviewer 1 Report

The manuscript entitled „Broad spectrum functional activity of structurally related monoanionic Au(III) bis(dithiolene) complexes“ was submitted by Y. L. Gal and co-workers to the journal “IJMS” in order to be considered for publication as an “Article”.

The manuscript is very comprehensive and deals with sixteen various gold(III) bis(dithiolene) and bis(diselenolene) complexes, though having similar structural moeties. In particular, their synthesis is reported, the cytotoxicity (complexes with Ph4P+ counter ion having stronger effects than Et4N+), investigations using the zebrafish model, antiplasmodial-antibacterial-antifungal-anti-HIV activity. They also performed studies on the inhibition of TrxR, interaction with DNA and albumin.

The authors are kindly asked to adjust the style of how references should be numbered in the running text, according to the author guidelines of the journal: “References should be numbered in order of appearance and indicated by a numeral or numerals in square brackets—e.g., [1] or [2,3], or [4–6].”

In general, the references in the Introduction would benefit from consideration of more recent work on gold metal complexes, reinforcing the current relevance of this important topic:

“numerous compounds of different transition metal ions are currently under development or reaching the clinical settings for medical applications.[i,ii]” There is even new or brand-new literature available, for example https://doi.org/10.1016/j.ccr.2021.214307 from 2022 about anticancer metal complexes or 10.1039/C9SC06460E from 2020 about antimicrobial metal complexes. Ver recently, there is a review on the rising interest in the development of metal complexes used for immunotherapy: https://doi.org/10.1002/asia.202200270 from 2022.

“In this context, gold complexes have gained much attention and endured extensive research in the medicinal chemistry domain for the treatment of cancer” There is also newer literature available than the cited work from 1996, for example: https://doi.org/10.1016/j.ccr.2009.02.019 from 2009, Nardon et al. Anticancer Res. 2014 34(1):487-92, among others. The authors are asked to consider newer references on that topic.

C^N)-cyclometalated ligands and N-heterocyclic carbenes (NHCs)” The authors are asked to consider more recent findings on gold-NHC complexes, such as M. Quintana Bioorg Med Chem from 2022, Q. Xiao Eur J Med Chem from 2022, E.A. Martynova Dalton Trans from 2022 C.M Gallati Daton Trans from 2021/2020, K.C Tong Front Chem from 2020, among others.

“The great stability of gold-NHC derivatives in biological media makes these complexes excellent potential chemotherapeutics” In my opinion, this statement cannot be made in such a general way, since gold complexes, for example, are also subject to ligand-exchange reactions, which influence the stability, especially in biological media:  10.1021/acs.inorgchem.1c00325 from 2021 (Reference xxvii). The authors are kindly asked to consider the vulnerability of gold complexes towards such exchange reactions.

“Nevertheless, a considerable number of gold(I) and gold(III) complexes, mostly complexes with a +1 oxidation state, have been tested against a broad spectrum of bacteria, fungi and parasites.” The authors are kindly asked to provide references serving as example, for instance https://doi.org/10.3389/fmicb.2022.846959 from 2022, 10.3389/fmicb.2022.815622 from 2022, or 10.1002/cmdc.202100381 from 2021.

Chart 1 and Figure 6 would benefit from a better resolution.

The authors are asked to include a short comment on the analytical characterization of the newly synthesized complexes in the manuscript when reporting on the synthesis (part 2.1).

I am wondering if it is really the right way how to present the charge of the anions?

The authors are asked to adjust the significant figures, i.e. two significant figures reported with the mean should also cause two significant figures with the standard deviation.

What was the reason not to test the effect of the gold complexes towards OVCAR8 after 24 h? The authors should provide a reason in their manuscript – or did I miss it? What was the effect of the complexes towards a non-cancer cell line, i.e. a regular human cell line (or did the authors therefore perform the zebrafish study?)?

Moreover, as the authors judged about Cisplatin in their introduction, it would be of interest to know the IC50 values of Cisplatin in the used cytotoxicity assay, just to compare and to evaluate if the current gold complexes can keep up with Cisplatin or might even exceed its effects? Moreover, it is suggested to incorporate established antibiotics/antimicrobials in their study as positive control (Table 4) – not only Auranofin which is not best suited to serve as positive control in this context.

“DNA-independent mechanism of some gold(III) complexes, while others report that binding to DNA is responsible for the cytotoxic effect” The authors are kindly asked to discuss recent findings that gold complexes interact with DNA G-quadruplexes, such as T. Rundstadler J Inorg Biochem from 2021, SMM Menches Chemistry from 2020 and J Inorg Biochem from 2020, D. Wragg Angew Chem Int Ed Engl from 2018.

“placing these compounds on the cut-ting edge of new research approaches in medicinal chemistry”. In order to allow a quick overview of the main findings regarding structure-activity-relationships and to allow other scientists working on the design of gold complexes to build on the findings of the current manuscript, the deducing of SAR would be nice. Which counter ion favors anticancer activity, which structural moieties are beneficial for low toxicity, what is the effect of the exo-cyclic atom? There are already some hints in the manuscript but including a table into the conclusions section summarizing the main findings would help the scientific community to grasp the outcomes of this very comprehensive study.

According to the author guidelines, the paragraph “Conclusions” should appear after the section “Materials and Methods”. Moreover, “Author Contribution” and “Conflicts of Interest” are missing as obvious referring to the template.

The authors are asked to provide CHN analysis for every complex since every complex was subject to biological testing (at least cytotoxicity using MTT). Therefore, purity must be confirmed.

Although there will be a close editing by the MDPI publisher, the authors are kindly asked to correct formal errors and inconsistency, such as: use of spaces, use of in-italics, different fonts, punctuation, subscript/superscript, typos, case shift, etc. Please carefully revise your manuscript concerning such issues.

All in all, the manuscript is generally suited to be published in the journal “IJMS”. However, and that is probably the major issue, the manuscript would benefit from the embedding in more recent work as suggested in the comments above. After incorporating these aspects, placing the study in the scientific surrounding, and correcting formal issues, the paper can be reconsidered for publication.

Author Response

Answers to Criticisms raised by Reviewer #1

Reviewer: The manuscript entitled „Broad spectrum functional activity of structurally related monoanionic Au(III) bis(dithiolene) complexes“ was submitted by Y. L. Gal and co-workers to the journal “IJMS” in order to be considered for publication as an “Article”.

The manuscript is very comprehensive and deals with sixteen various gold(III) bis(dithiolene) and bis(diselenolene) complexes, though having similar structural moeties. In particular, their synthesis is reported, the cytotoxicity (complexes with Ph4P+ counter ion having stronger effects than Et4N+), investigations using the zebrafish model, antiplasmodial-antibacterial-antifungal-anti-HIV activity. They also performed studies on the inhibition of TrxR, interaction with DNA and albumin.

The authors are kindly asked to adjust the style of how references should be numbered in the running text, according to the author guidelines of the journal: “References should be numbered in order of appearance and indicated by a numeral or numerals in square brackets—e.g., [1] or [2,3], or [4–6].”

Answer: We kindly appreciate the reviewers´comments. We have modified the style of the references. They are now all numbered with numerals in square brackets.

Reviewer: In general, the references in the Introduction would benefit from consideration of more recent work on gold metal complexes, reinforcing the current relevance of this important topic:

“numerous compounds of different transition metal ions are currently under development or reaching the clinical settings for medical applications.[i,ii]” There is even new or brand-new literature available, for example https://doi.org/10.1016/j.ccr.2021.214307 from 2022 about anticancer metal complexes or 10.1039/C9SC06460E from 2020 about antimicrobial metal complexes. Ver recently, there is a review on the rising interest in the development of metal complexes used for immunotherapy: https://doi.org/10.1002/asia.202200270 from 2022.

Answer: We sincerely appreciate the suggestion and all the three suggested references were introduced in the manuscript.

Reviewer: “In this context, gold complexes have gained much attention and endured extensive research in the edicinal chemistry domain for the treatment of cancer” There is also newer literature available than the cited work from 1996, for example: https://doi.org/10.1016/j.ccr.2009.02.019 from 2009, Nardon et al. Anticancer Res. 2014 34(1):487-92, among others. The authors are asked to consider newer references on that topic.

Answer: Thanks for the suggestion. Reference 11 was replaced by C. Nardon, G. Boscutti, D. Fregona, Beyond platinum: gold complexes as anticancer agents, Anticancer Res. 34 (2014) 487-492.

Reference 13 was replaced by L. Kou, S. Wei, P. Kou, Current progress and perspectives on using gold compounds for the modulation of tumor cell metabolism, Front. Chem 9 (2021) 733463.

Reviewer:  “C^N)-cyclometalated ligands and N-heterocyclic carbenes (NHCs)” The authors are asked to consider more recent findings on gold-NHC complexes, such as M. Quintana Bioorg Med Chem 2022, Q. Xiao Eur J Med Chem from 2022, E.A. Martynova Dalton Trans from 2022 C.M Gallati Daton Trans from 2021/2020, K.C Tong Front Chem from 2020, among others.

“The great stability of gold-NHC derivatives in biological media makes these complexes excellent potential chemotherapeutics” In my opinion, this statement cannot be made in such a general way, since gold complexes, for example, are also subject to ligand-exchange reactions, which influence the stability, especially in biological media:  10.1021/acs.inorgchem.1c00325 from 2021 (Reference xxvii). The authors are kindly asked to consider the vulnerability of gold complexes towards such exchange reactions.

“Nevertheless, a considerable number of gold(I) and gold(III) complexes, mostly complexes with a +1 oxidation state, have been tested against a broad spectrum of bacteria, fungi and parasites.” The authors are kindly asked to provide references serving as example, for instance https://doi.org/10.3389/fmicb.2022.846959 from 2022, 10.3389/fmicb.2022.815622 from 2022, or 10.1002/cmdc.202100381 from 2021.

Answer: We kindly appreciate the suggestions. The suggested publications on more recent findings were included in the revised version of the manuscript whenever appropriate.

Reviewer: Chart 1 and Figure 6 would benefit from a better resolution.

Answer: We apologize for the poor quality of previous Chart 1 and Fig. 6. As requested, Chart 1  and Fig 6 were modified and their resolution improved in the revised version.

Reviewer: The authors are asked to include a short comment on the analytical characterization of the newly synthesized complexes in the manuscript when reporting on the synthesis (part 2.1).

Answer: We kindly appreciate the suggestion. Accordingly, we added the following: After recrystallization in acetonitrile, the gold complexes are isolated as dark greenish crystalline solids. The newly synthesized complexes were all characterized using 1H NMR, HRMS, elemental analysis and cyclic voltammetry (see top of page 4, highlighted in yellow).

Reviewer: I am wondering if it is really the right way how to present the charge of the anions?

Answer: We kindly appreciate the comment. We presume that the question is related to scheme 2. In this scheme we presented the complex between brackets and the charge as superscript. Actually, for metal bis(dithiolene) complexes in general, the ligand, dithiolene, is a non-innocent ligand and as such due to electron delocalization among the metallacycle the oxidation state of the metal and the one of the ligand are unclear. Therefore, this writing gives the overall charge of the complex.

Question: The authors are asked to adjust the significant figures, i.e. two significant figures reported with the mean should also cause two significant figures with the standard deviation.

Answer: We kindly appreciate the comment and have made the adequate corrections in the revised version as suggested.

Question: What was the reason not to test the effect of the gold complexes towards OVCAR8 after 24 h? The authors should provide a reason in their manuscript – or did I miss it? What was the effect of the complexes towards a non-cancer cell line, i.e. a regular human cell line (or did the authors therefore perform the zebrafish study?)?

Answer: We kindly appreciate the comment. OVCAR8 are ovarian cisplatin-resistant cells. They were introduced to ascertain the ability of the complexes to overcome cisplatin resistance, since the majority of ovarian tumors eventually recur in a drug resistant form. They were used as a control cell line, selecting the incubation time point of 48h. Authors in a previous publication on a series of Gold(III) dithiolenes (please see ref. 47) introduced a non-cancer cell line, the V79 fibroblasts to evaluate complexes selectivity for cancer cells. Therefore, in this paper we used the zebrafish embryo model as a step ahead model to predict toxicity and as a bridge between in vitro assays and in vivo studies.

Question: Moreover, as the authors judged about Cisplatin in their introduction, it would be of interest to know the IC50 values of Cisplatin in the used cytotoxicity assay, just to compare and to evaluate if the current gold complexes can keep up with Cisplatin or might even exceed its effects? Moreover, it is suggested to incorporate established antibiotics/antimicrobials in their study as positive control (Table 4) – not only Auranofin which is not best suited to serve as positive control in this context.

Answer: We kindly appreciate the criticism. In our previous published paper (please see ref. 47) the activity of cisplatin against the A2780 Cisplatin sensitive cells was already evaluated. After 48h incubation, the IC50 value was 3.6±1.3 which was higher than the IC50 values found for the majority of the complexes. Therefore, we have added the IC50 values found for Cisplatin at 24h in the A2780 cells and 48h in the OVCAR8 to in table 2.

Question: “DNA-independent mechanism of some gold(III) complexes, while others report that binding to DNA is responsible for the cytotoxic effect” The authors are kindly asked to discuss recent findings that gold complexes interact with DNA G-quadruplexes, such as T. Rundstadler J Inorg Biochem from 2021, SMM Menches Chemistry from 2020 and J Inorg Biochem from 2020, D. Wragg Angew Chem Int Ed Engl from 2018.

Answer: We kindly appreciate the suggestion. Therefore, additional references and a brief discussion was introduced according to the reviewer suggestions.

Question: “placing these compounds on the cut-ting edge of new research approaches in medicinal chemistry”. In order to allow a quick overview of the main findings regarding structure-activity-relationships and to allow other scientists working on the design of gold complexes to build on the findings of the current manuscript, the deducing of SAR would be nice. Which counter ion favors anticancer activity, which structural moieties are beneficial for low toxicity, what is the effect of the exo-cyclic atom? There are already some hints in the manuscript but including a table into the conclusions section summarizing the main findings would help the scientific community to grasp the outcomes of this very comprehensive study.

Answer: We kindly appreciate the suggestion. However, we need to be modest to draw solid conclusions regarding SARs. Although the complexes showed promising activities as anticancer and antimicrobial agents. In general complexes having Ph4P+ as counter-ion are more cytotoxic to the cancer cells, less toxic to zebrafish embryos and also present higher antiplasmodial activity. By its turn, complexes having Et4N+ as counter-ion have activity for Gram positive S. aureus, and the Candida species C. glabrata. In our opinion, further studies are needed with more related compounds to draw conclusions on a SARs basis. Nevertheless, these main findings are summarized in the Conclusions part of the revised manuscript, highlighted in yellow.

Question: According to the author guidelines, the paragraph “Conclusions” should appear after the section “Materials and Methods”. Moreover, “Author Contribution” and “Conflicts of Interest” are missing as obvious referring to the template.

Answer: We thank the reviewer for these corrections. Therefore, the conclusion section has been moved after the Materials and methods section. Author contributions were also added. It now reads: Y.L.G. performed the synthesis and the redox analysis of the gold bis(dithiolene) complexes, A.F.F. prepared and analyzed the gold bis(diselenolene) complexes, O.J., T.R. and V.D. performed the X-ray data collections and the structure refinements, D. L. conceived, designed the complexes, coordinated the chemical section and wrote the article together with F.M., D.Fo, D.Fr, M.P. perfomed the antiplasmodial activity studies, M.M. performed the toxicity studies in zebrafish embryo, C.S., S.A.S, J.H.L. perfomed the antibacterial and antifungal activity studies, I.B., N.T. performed the anti-HIV activity studies, T.S.M. performed the studies of interaction with HAS, J.F.G. performed DNA electrophoresis, F.M. performed the mechanistic studies and coordinate the biological section.

The conflict of interest section was also added

Question: The authors are asked to provide CHN analysis for every complex since every complex was subject to biological testing (at least cytotoxicity using MTT). Therefore, purity must be confirmed.

Answer: Thanks for the suggestion. The C, H, N, analysis was provided. See the revised 3.2. General synthetic procedures, where the modifications are highlighted in yellow.

Although there will be a close editing by the MDPI publisher, the authors are kindly asked to correct formal errors and inconsistency, such as: use of spaces, use of in-italics, different fonts, punctuation, subscript/superscript, typos, case shift, etc. Please carefully revise your manuscript concerning such issues.

Answer: We kindly appreciate the suggestion. As requested, a thorough revision was made to accommodate the above-mentioned issues.

Question: All in all, the manuscript is generally suited to be published in the journal “IJMS”. However, and that is probably the major issue, the manuscript would benefit from the embedding in more recent work as suggested in the comments above. After incorporating these aspects, placing the study in the scientific surrounding, and correcting formal issues, the paper can be reconsidered for publication.

Answer: We thank the reviewer for his/her suggestions to improve our MS. We add additional references on the relevant topics and more recent work, whenever appropriate and improve the Conclusions in order to highlight the most important findings of our work. The whole manuscript was also revised concerning minor issues like punctuation, talics, and other minor issues.

Reviewer 2 Report

Manuscript

The article entitled “Broad spectrum functional activity of structurally related

monoanionic Au(III) bis(dithiolene) complexes” by Fernanda Marques and co-workers reports, the  anticancer, antimicrobial activities and   anti-HIV activity of  structurally-related monoanionic gold(III) bis(dithiolene/diselenolene) complexes, differing by the nature of  the heteroatom linked to gold atom, of the nitrogen  substituent of the thiazoline ring and the nature of the exocyclic atom or group  of atoms and of the counter-ion. Although the work was developed with expertise covering different biological potentialities  there are some aspects that the authors must reformulate to facilitate the reading.

i)               The authors must check the references since they are indicated by letters and not by numbers.

ii)             No reference to Chart 1 that appears before the results and discussion.

iii)     Although I understand the efforts of the authors to compile all the synthetic work in scheme 1 and chart 1, in my opinion, as it is, requires some  efforts to understand the structures of the complexes prepared. Why not to be more conventional and to separate the scheme in two parts showing clearly the access to the dithiolene (X = S) and to  diselenolene (X = Se) complexes, and what is the R and the Y of  those structures, associated to the abbreviation  that are going to be used in the discussion of the results obtained.  I am sure the readers would be much happier to follow the remaining discussion of the other results.  

iv)    Is better to refer sodium methanolate as sodium methoxide

v) In some comments concerning the discussion of biological results I suggest

the authors to add the values presented in table in order to facilitate the reading

without coming back to the table. Like in this: This is even more obvious when comparing the same gold complexes [N][AuSEt(C=S)] vs [P][AuSEt(C=S)], [N][AuSeEt(C=Se)] vs [P][AuSeEt(C=Se)] and  [N][AuSeiPr(C=S)] vs [P][AuSeiPr(C=S)]. In the last part of the discussion  the authors were able to do this  and it is much easier to read.

vi)            In the NMR the letter J is in italic and in the indication of the molecular ion the +. must be indicated as Superscript. The authors should  add the NMR and the mass spectra of the compounds to SI.

vii)          Section 4.4.2. and others check some concentrations since instead of point the authors use comma.   What means the s after  5 × 105 CFU/mL s?

viii)        Why the complex [N][AuSEt(=C(CN)2)] has no microanalysis?

Author Response

Answers to criticisms raised by Reviewer #2

Reviewer: The article entitled “Broad spectrum functional activity of structurally related monoanionic Au(III) bis(dithiolene) complexes” by Fernanda Marques and co-workers reports, the anticancer, antimicrobial activities and anti-HIV activity of structurally-related monoanionic gold(III) bis(dithiolene/diselenolene) complexes, differing by the nature of the heteroatom linked to gold atom, of the nitrogen  substituent of the thiazoline ring and the nature of the exocyclic atom or group of atoms and of the counter-ion. Although the work was developed with expertise covering different biological potentialities there are some aspects that the authors must reformulate to facilitate the reading.

i)The authors must check the references since they are indicated by letters and not by numbers.

Answer: We kindly appreciate the comment. This was modified, the references are now indicated by numbers.

Reviewer: ii) No reference to Chart 1 that appears before the results and discussion.

Answer: We appreciate the comment. Chart 1 was already referred in the paragraph mentioning the investigated compounds, in the Introduction section

Reviewer: iii) Although I understand the efforts of the authors to compile all the synthetic work in scheme 1 and chart 1, in my opinion, as it is, requires some efforts to understand the structures of the complexes prepared. Why not to be more conventional and to separate the scheme in two parts showing clearly the access to the dithiolene (X = S) and to diselenolene (X = Se) complexes, and what is the R and the Y of  those structures, associated to the abbreviation that are going to be used in the discussion of the results obtained. I am sure the readers would be much happier to follow the remaining discussion of the other results.

Answer: We kindly acknowledge the comment. Scheme 1 has been modified and now shows the two synthetic routes, the one for the gold bis(dithiolene) and the one for the gold bis(diselenolene). In the scheme caption the nature of C, R and Y for both families has been specified

Question: iv) Is better to refer sodium methanolate as sodium methoxide

Answer: We kindly appreciate the suggestion and Sodium methanolate was replaced by sodium methoxide

Question: v) In some comments concerning the discussion of biological results I suggest the authors to add the values presented in table in order to facilitate the reading without coming back to the table. Like in this: This is even more obvious when comparing the same gold complexes [N][AuSEt(C=S)] vs [P][AuSEt(C=S)], [N][AuSeEt(C=Se)] vs [P][AuSeEt(C=Se)] and  [N][AuSeiPr(C=S)] vs [P][AuSeiPr(C=S)]. In the last part of the discussion, the authors were able to do this and it is much easier to read.

Answer: We kindly appreciate the suggestion. Accordingly, the values presented in Table 2 were added according to reviewer suggestion.

Question: vi) In the NMR the letter J is in italic and in the indication of the molecular ion the +. must be indicated as Superscript. The authors should add the NMR and the mass spectra of the compounds to SI.

Answer: Thanks for the suggestions and observation. Concerning the NMR, J is now in italic. For the HRMS study our notation is correct for the complexes as they measure ionization of cluster [2C+,complex-]+ thus in this case the ion is positive. We added the charge as superscript as requested. Concerning the precursors we modified and added M+• as requested. We also added all the 1H NMR of the reported complexes in the supporting information file. 

Question vii) Section 4.4.2. and others check some concentrations since instead of point the authors use comma. What means the s after 5 × 105 CFU/mL s?

Answer: We kindly appreciate the comment. Our mistake, corrected in the revised version.

Question: viii) Why the complex [N][AuSEt(=C(CN)2)] has no microanalysis?

Answer: We acknowledge the comment, and accordingly the [N][AuSEt(=C(CN)2)] analysis was added.

We thank the reviewer for his/her comments to improve our manuscript.

Reviewer 3 Report

This paper describes synthesis, characterization, and biological studies (e.g. IC50) for many new gold complexes having S-containing ligands.

From the viewpoints of chemistry and biology, is must be worth publishing in IJMS almost as it is, because experimental data and description did not contain serious issues.

However, some description comparing with known drug metal complexes such as Cisplatin and Auranofin may mislead the readers' understanding. Because Cisplatin's antitumor mechanism is binding DNA to change DNA's steric structure, while Auranofin is a typical example of antirheumatic drugs.

The present study mentioned redox properties as a sole Au complex may differs from Cisplatin (having importance of substitution of ligand and solubility) or Auranofin (not antitumor but antirheumatic drug).

What aspects of such famous metal complex drugs is similar to the present Au complexes should be stated clearly when compared in text.

Please add such explanation before acceptance.

That's all.

Author Response

Answers to criticisms raised by Reviewer #3

Reviewer: This paper describes synthesis, characterization, and biological studies (e.g. IC50) for many new gold complexes having S-containing ligands.

From the viewpoints of chemistry and biology, is must be worth publishing in IJMS almost as it is, because experimental data and description did not contain serious issues.

However, some description comparing with known drug metal complexes such as Cisplatin and Auranofin may mislead the readers' understanding. Because Cisplatin's antitumor mechanism is binding DNA to change DNA's steric structure, while Auranofin is a typical example of antirheumatic drugs.

The present study mentioned redox properties as a sole Au complex may differs from Cisplatin (having importance of substitution of ligand and solubility) or Auranofin (not antitumor but antirheumatic drug).

What aspects of such famous metal complex drugs is similar to the present Au complexes should be stated clearly when compared in text.

Please add such explanation before acceptance.

Answer: We kindly appreciate the general positive comments of the reviewer.

In the Conclusion Section, a brief summary of the mechanism of action of both Cisplatin and Auronofin was presented and a tentative description of the similarities with the mechanism of action of the gold complexes.

We investigated the redox properties of Auranofin using the same experimental conditions used for all the gold bis(dithiolene) complexes and bis(diselenolene) complexes. Concerning Cisplatin, this study was not possible in the same conditions due to the insolubility of the Cisplatin. We added in the text :

“For comparison, we also investigated the redox behavior of Auranofin using the same experimental conditions in CH2Cl2 in the presence of NBu4PF6 as supporting electrolyte. Using these conditions, upon anodic scan only one irreversible oxidation process is observed at 1.42 V vs SCE. Therefore all these monoanionic gold bis(dithiolene) and bis(diselenolene) complexes are easier to oxidize than Auranofin.”

We thank the reviewer for his/her support and recommendations.

Reviewer 4 Report

The manuscript from Le Gal et al. reports several biological properties of a series of gold(III) complexes. The results are interesting, well presented and support the numerous literature data about gold complexes with different ligands. I recommend the publication of this manuscript after minor changes.

 1) Assign a number to each complex, in order to make easier the reading and understanding of the text, figures and tables.

2) Page 11, Auranofin IC50 SD value for S.aureus is missing

3) The conclusion section needs to be improved, for instance highlighting the future perspective and reporting the novelty of the presented outcomes.

4) Check for grammar and typos (verbs, pronouns and so on) all along the manuscript.

Author Response

Answers to criticisms raised by Reviewer #4

Reviewer: The manuscript from Le Gal et al. reports several biological properties of a series of gold(III) complexes. The results are interesting, well presented and support the numerous literature data about gold complexes with different ligands. I recommend the publication of this manuscript after minor changes.

  • Assign a number to each complex, in order to make easier the reading and understanding of the text, figures and tables.

Answer: We kindly appreciate the suggestion. However, such numbering of complexes would raise some problems. In the first place, it seems simple, but each time you need to refer to a table to find out if it is a dithiolene or a diselenolene, to check the substituent on the nitrogen atom and to find the exocyclic substituent, the numbering would not help at all. Our notation allows a fast identification of all these parameters at once especially in the text for comparison purposes. We have therefore kept the notation used.

Reviewer:

  • Page 11, Auranofin IC50 SD value for S.aureus is missing.

Answer: We kindly appreciate the comment. This issue is now corrected in the revised version.

Reviewer:

  • The conclusion section needs to be improved, for instance highlighting the future perspective and reporting the novelty of the presented outcomes.

Answer: We kindly appreciate the comment and therefore the Conclusion section was improved according to reviewer suggestions.

Reviewer:

4) Check for grammar and typos (verbs, pronouns and so on) all along the manuscript.

Answer: We kindly appreciate the suggestion. We have performed a thorough revision of the whole manuscript.

We thank the reviewer for his/her support and recommendations.

Round 2

Reviewer 1 Report

The authors Y. L. Gal and co-workers submitted a revised version of their manuscript. The authors have made an effort to implement the suggestions for improvement. The authors have also corrected most of the suggestions or provided comprehensible rebuttals. Nevertheless, I would like to comment on the following minor aspects.

The question why the effect of the gold complexes towards OVCAR8 was not tested after 24 h (!) is not answered exactly. Well, this does not significantly affect the quality of the manuscript.

The suggestion “Moreover, it is suggested to incorporate established antibiotics/antimicrobials in their study as positive control (Table 4) – not only Auranofin which is not best suited to serve as positive control in this context.” was not addressed at all, neither by the submission of additional data nor by a discussion in the letter to the reviewers.

The authors state that they corrected formal errors and inconsistency. However, there are still such errors in the manuscript, e.g., consistent use of spaces / use of in-italics, different fonts, case shift. Please - once again - carefully revise your manuscript concerning such issues.

After this minor revision, I support further processing of the manuscript for publication. All the best!

Author Response

Reviewer#1

Comments and Suggestions for Authors

Reviewer: The authors Y. L. Gal and co-workers submitted a revised version of their manuscript. The authors have made an effort to implement the suggestions for improvement. The authors have also corrected most of the suggestions or provided comprehensible rebuttals. Nevertheless, I would like to comment on the following minor aspects.

The question why the effect of the gold complexes towards OVCAR8 was not tested after 24 h (!) is not answered exactly. Well, this does not significantly affect the quality of the manuscript.

Answer: We kindly appreciate the comment. OVCAR8 are ovarian cisplatin-resistant cells. They were introduced to ascertain the ability of the complexes to overcome cisplatin resistance, since the majority of ovarian tumors display chemoresistance to cisplatin-based chemotherapy. In addition, to compare compound´s activity with the cisplatin-sensitive A2780. The comparative study was performed at 48h and indicate that at this incubation time most of the complexes presented higher activity than cisplatin.

Concening the effect of complexes towards a non-cancer cell line, Authors in a previous publication on a series of Gold (III) tithiolenes (please see ref. 47) introduced a non-cancer cell line, the V79 fibroblasts to evaluate the selectivity of complexes for cancer cells. Therefore, in this paper we used the zebrafish embryo model as a step ahead model to predict toxicity and as a bridge between in vitro assays and in vivo studies.

In order to be clearer for the reader, we have introduced the following modifications (highlighted in green) to the first paragraph in page 7:

The anticancer activity of the sixteen complexes under study and the reference drugs Cisplatin and Auranofin, was evaluated in the A2780 cisplatin-sensitive ovarian cancer cells at 24 and 48h incubation. The Cisplatin-resistant OVCAR8 cells were introduced for comparison at 48h incubation, to ascertain the ability of the complexes to overcome cisplatin resistance, since the majority of ovarian tumors eventually recurs in a drug resistant form.

Reviewer: The suggestion “Moreover, it is suggested to incorporate established antibiotics/antimicrobials in their study as positive control (Table 4) – not only Auranofin which is not best suited to serve as positive control in this context.” was not addressed at all, neither by the submission of additional data nor by a discussion in the letter to the reviewers.

Answer: We appreciate the comment and apologize for the lack of response, our fault. Indeed, Auranofin is the only commercially available compound structurally related to the complexes under study, and that has been used as antimicrobial. In our view it makes no sense to introduce currently in use antimicrobials which have nothing to do with gold complexes in terms of structure or possible mechanisms of action.

Nevertheless, and to make this clearer to the reader, we have slightly modified the first paragraph in page 11, which now reads as follows:

2.6. Antibacterial and antifungal activity

The antibacterial and antifungal activity of the complexes under study was assessed based on their MIC values against E. coli, S. aureus, C. albicans and C. glabrata. Auranofin was used in the present study as positive control since it is the only commercially available antimicrobial compound structurally related to the complexes under study. Results obtained are presented in Table 4.

Reviewer: The authors state that they corrected formal errors and inconsistency. However, there are still such errors in the manuscript, e.g., consistent use of spaces / use of in-italics, different fonts, case shift. Please - once again - carefully revise your manuscript concerning such issues.

Answer: Thanks for the indication. A second round of revisions was carried out. Hope now those minor formal errors are cleaned!

Reviewer: After this minor revision, I support further processing of the manuscript for publication. All the best!

Answer: We want to express our gratitude for the high quality of the revisions performed and the opportunity to increase the quality of the final manuscript. Best wishes!

Reviewer 2 Report

The authors have in account my concerns and in my opinion this excellent article is in conditions to be accepted.

Author Response

Reviewer2 comment:

The authors have in account my concerns and in my opinion this excellent article is in conditions to be accepted.

Answer: We kindly appreciate the comment! Best wishes!